# Dimerisation of the PICTS complex via LC8/Cut-up drives co-transcriptional transposon silencing in *Drosophila*

**Evelyn L Eastwood[1], Kayla A Jara[2†], Susanne Bornelöv[1†], Marzia Munafo[1], Vasileios Frantzis[1], Emma Kneuss[1], Elisar J Barbar[2], Benjamin Czech[1]*, Gregory J Hannon[1]***

[1]Cancer Research UK Cambridge Institute, University of Cambridge, Li Ka Shing Centre, Cambridge, United Kingdom; [2]Department of Biochemistry and Biophysics, Oregon State University, Corvallis, United States

**Abstract** In animal gonads, the PIWI-interacting RNA (piRNA) pathway guards genome integrity in part through the co-transcriptional gene silencing of transposon insertions. In *Drosophila* ovaries, piRNA-loaded Piwi detects nascent transposon transcripts and instructs heterochromatin formation through the *Panoramix-induced co-transcriptional silencing* (PICTS) complex, containing Panoramix, Nxf2 and Nxt1. Here, we report that the highly conserved dynein light chain LC8/Cut-up (Ctp) is an essential component of the PICTS complex. Loss of Ctp results in transposon de-repression and a reduction in repressive chromatin marks specifically at transposon loci. In turn, Ctp can enforce transcriptional silencing when artificially recruited to RNA and DNA reporters. We show that Ctp drives dimerisation of the PICTS complex through its interaction with conserved motifs within Panoramix. Artificial dimerisation of Panoramix bypasses the necessity for its interaction with Ctp, demonstrating that conscription of a protein from a ubiquitous cellular machinery has fulfilled a fundamental requirement for a transposon silencing complex.

**\*For correspondence:**
benjamin.czech@cruk.cam.ac.uk
(BC);
greg.hannon@cruk.cam.ac.uk
(GJH)

[†]These authors contributed equally to this work

**Competing interests:** The authors declare that no competing interests exist.

## Introduction

Large portions of eukaryotic genomes are occupied by repetitive sequences and mobile genetic elements (*Cosby et al., 2019*). The de novo establishment of heterochromatin at these regions prevents transposon mobilisation and aberrant recombination events, and the ability of small RNAs to provide sequence specificity to this mechanism has emerged as a fundamental element of eukaryotic genome defence systems (*Martienssen and Moazed, 2015*). Specifically, nuclear Argonaute proteins bound to small RNA partners recognise nascent RNA transcripts and trigger a molecular cascade that leads to recruitment of epigenetic modifiers to the target region (*Czech et al., 2018*; *Martienssen and Moazed, 2015*; *Ozata et al., 2019*). This co-transcriptional process ultimately results in the formation of a repressive chromatin environment at the target locus, with most of our understanding originating from studies in yeast and plants (*Martienssen and Moazed, 2015*).

In animal gonads, the piRNA pathway represses mobile element activity, in part, through the co-transcriptional gene silencing (TGS) of transposon insertions (*Czech et al., 2018*; *Ozata et al., 2019*). In *Drosophila melanogaster*, where this pathway has been extensively studied, piRNA-guided TGS depends on the PIWI-clade Argonaute protein Piwi (*Klenov et al., 2014*; *Le Thomas et al., 2013*; *Rozhkov et al., 2013*; *Sienski et al., 2012*; *Wang and Elgin, 2011*). Loci targeted by Piwi are associated with repressive chromatin states marked by the absence of di-methylated H3K4 (H3K4me2) and the presence of di- and tri-methylated H3K9 (H3K9me2/3). These regions are also coated with Heterochromatin Protein 1a (HP1a/Su(var)205) (*Wang and Elgin, 2011*). Loss of Piwi results in marked de-repression of transposon mRNA levels which is coupled with loss of repressive

histone modifications and increased occupancy of RNA polymerase II at transposon insertions (*Klenov et al., 2014*; *Le Thomas et al., 2013*; *Rozhkov et al., 2013*; *Sienski et al., 2012*; *Wang and Elgin, 2011*). Transposon control at the transcriptional level through nuclear PIWI proteins is not unique to *Drosophila*. In mouse embryonic testes, the nuclear PIWI protein MIWI2 directs TGS to evolutionarily young transposon insertions, which are silenced in a process that alters both the histone modification and DNA methylation landscapes (*Aravin et al., 2008*; *Carmell et al., 2007*; *Molaro et al., 2014*; *Pezic et al., 2014*). While the mechanism of TGS in mammals is not fully understood, recent work linked two nuclear partners of MIWI2, TEX15 and SPOCD1, to this process (*Schöpp et al., 2020*; *Zoch et al., 2020*).

The molecular mechanism linking target recognition by Piwi with transcriptional repression is yet to be fully resolved. Removal of H3K4 methylation marks depends on the lysine demethylase Lsd1/Su(var)3–3, assisted by CoRest, while H3K9me3 is deposited by the histone methyltransferase dSetDB1/Eggless (Egg) and its cofactor Windei (Wde) (*Czech et al., 2013*; *Handler et al., 2013*; *Koch et al., 2009*; *Muerdter et al., 2013*; *Osumi et al., 2019*; *Rangan et al., 2011*; *Sienski et al., 2015*; *Yu et al., 2015*). The chromatin-associated protein Ovaries absent (Ova), which bridges HP1a and Lsd1, the SUMO ligase Su(var)2–10, the nucleosome remodeller Mi-2 and the histone deacetylase Rpd3 additionally contribute to the heterochromatin formation process (*Mugat et al., 2020*; *Ninova et al., 2020*; *Yang et al., 2019*). Loss of any of these factors in ovaries impairs TGS and transposon repression, yet how they are recruited to transposon loci and cooperate in target repression remains elusive (*Czech et al., 2013*; *Handler et al., 2013*; *Muerdter et al., 2013*; *Mugat et al., 2020*; *Ninova et al., 2020*; *Sienski et al., 2015*; *Yang et al., 2019*; *Yu et al., 2015*).

Recent work has uncovered a complex consisting of Panoramix (Panx), Nuclear export factor 2 (Nxf2) and NTF2-related export protein 1 (Nxt1) termed PICTS (also known as SFiNX) that functions downstream of Piwi and upstream of the chromatin modifying machinery to effect TGS (*Batki et al., 2019*; *Fabry et al., 2019*; *Murano et al., 2019*; *Sienski et al., 2015*; *Yu et al., 2015*; *Zhao et al., 2019*). While both Panx and Nxf2 can induce transcriptional repression when artificially recruited to reporters, the link to epigenetic modifiers and the transcriptional silencing activity of the PICTS complex seems to reside within Panx itself (*Batki et al., 2019*; *Fabry et al., 2019*).

Here, we report that the highly conserved dynein light chain LC8/Cut-up (Ctp) is a critical TGS factor and an integral part of the PICTS complex. Ctp is essential for transposon repression in both the somatic and germline compartments of *Drosophila* ovaries and, like Panx and Nxf2, can induce transcriptional silencing when artificially tethered to a reporter locus. In contrast to Panx and Nxf2, whose functions are specific to transposon control (*Batki et al., 2019*; *Fabry et al., 2019*; *Murano et al., 2019*; *Sienski et al., 2015*; *Yu et al., 2015*; *Zhao et al., 2019*), Ctp is a ubiquitously expressed and essential protein with wide-ranging cellular functions (*Barbar, 2008*; *Jespersen and Barbar, 2020*; *Rapali et al., 2011b*). While first identified as a component of the dynein motor complex (*King and Patel-King, 1995*), Ctp, which itself forms a homodimer, has since been shown to function as a dimerisation hub protein, promoting the assembly and stabilisation of many dynein-independent protein complexes (*Barbar and Nyarko, 2014*; *Jespersen and Barbar, 2020*). We show that Ctp promotes dimerisation of the PICTS complex through its interaction with two conserved motifs in Panx. In the absence of Ctp, Panx fails to self-associate and is unable to support transposon silencing. Considered together, our data reveal that Ctp-mediated higher order PICTS complex assembly is essential for heterochromatin formation and silencing.

## Results

### Ctp associates with the PICTS complex and is required for transposon control

Previously, we and others showed that Panx, Nxf2, and Nxt1 form a complex that acts downstream of Piwi target engagement to silence transposons at the transcriptional level (*Batki et al., 2019*; *Fabry et al., 2019*; *Murano et al., 2019*; *Zhao et al., 2019*). As part of this study, we immunoprecipitated GFP-tagged Panx and Nxf2 from ovary lysates and identified the associated proteins by mass spectrometry (*Fabry et al., 2019*). As well as known components of the PICTS complex, the protein Cut-up (Ctp) was significantly enriched in both experiments (*Figure 1A*). This was an unexpected result, since Ctp is a member of the LC8 family of dynein light chains and a component of

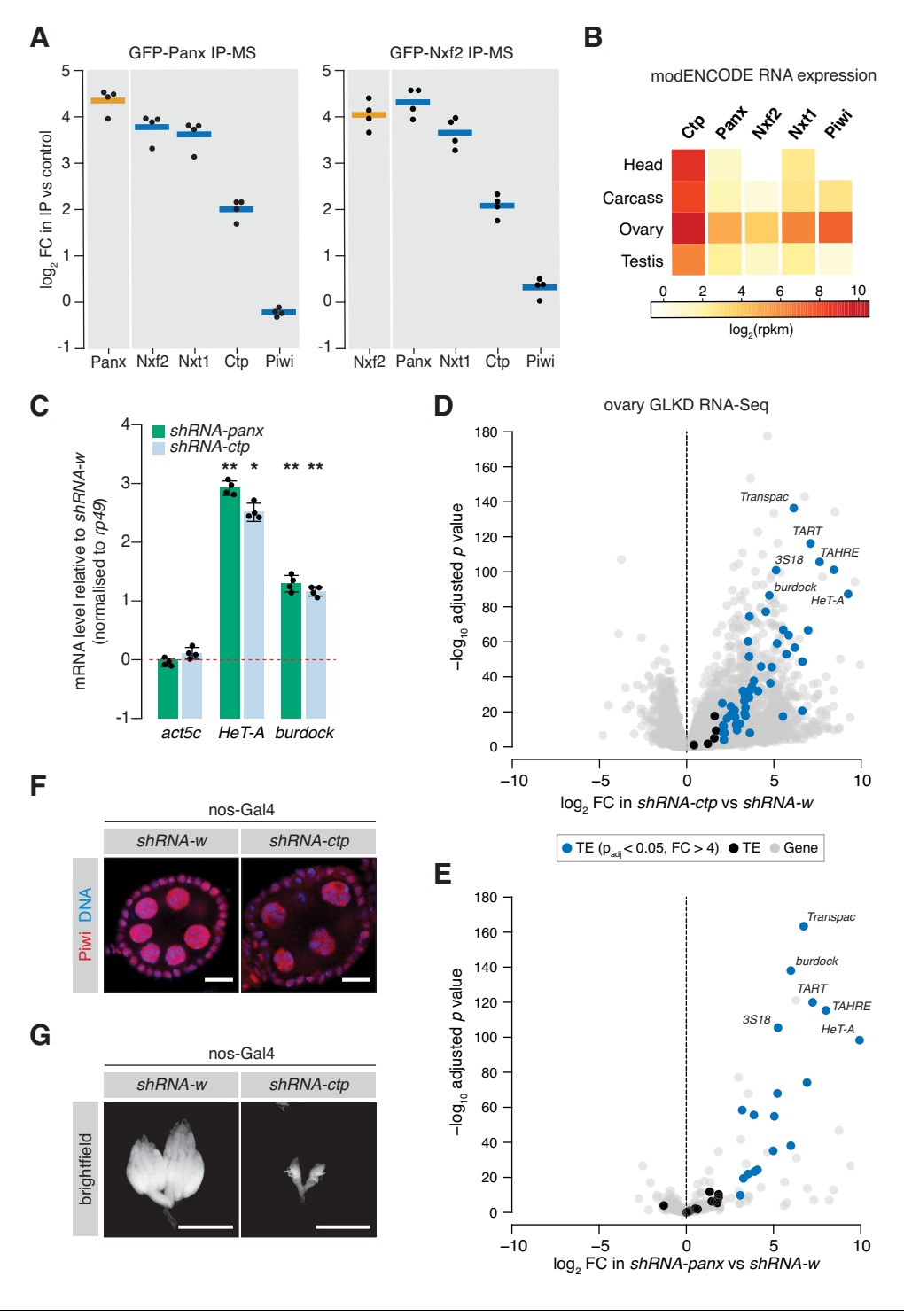

**Figure 1.** Ctp associates with the PICTS complex and is required for transposon silencing in the fly germline. (**A**) Enrichment plot showing the fold-change of PICTS complex proteins and Piwi in the GFP-Panx and GFP-Nxf2 IPs vs the control (n = 4). (**B**) Heatmap showing expression levels of *ctp, cdlc2, panx, nxf2, nxt1*, and *piwi* in various *Drosophila* tissues. (**C**) Bar graph showing fold-changes in ovary RNA levels of the house-keeping gene *act5c* and germline-specific transposons *HeT-A* and *burdock* upon germline knockdown of the indicated gene as measured by qPCR. (*) p<0.05 (**) p<0.001 (unpaired t-test). Error bars indicate standard deviation (n = 4). (**D**) Volcano plot showing fold-change and significance (adjusted p value) of genes and transposons between *ctp* GLKD and control as measured by RNA-seq. TE, transposable element. (**E**) As in D but comparing *panx* GLKD with control. (**F**) *Figure 1 continued on next page*

*Figure 1 continued*

Immunofluorescence images showing Piwi localisation in ovaries upon knockdown of the indicated gene. Scale bar = 10 µm. (G) Images showing ovary morphology visualised by DAPI staining upon knockdown of the indicated gene. Scale bar = 1 mm.

The online version of this article includes the following source data and figure supplement(s) for figure 1:

**Source data 1.** Differential expression analysis of ovary RNA-seq data used for volcano plots shown in *Figure 1D* and *Figure 1E*.

**Figure supplement 1.** Ctp is highly conserved and essential for germline transposon repression.

the dynein motor complex. Moreover, in contrast to Piwi, Panx, and Nxf2, which display ovary-enriched expression, Ctp is ubiquitously expressed (*Figure 1B*) and essential for viability, with loss-of-function mutations in *Drosophila* causing embryonic lethality (*Dick et al., 1996*). The LC8 family of dynein light chains was first identified as a component of cytoplasmic dynein (*King and Patel-King, 1995*) and this family is extremely well conserved, with *Drosophila* Ctp differing by only four amino acids from its mammalian ortholog (*Figure 1—figure supplement 1A*). These proteins have since been shown to be involved in a number of dynein-independent processes, including transcriptional regulation, mitotic spindle positioning, and apoptosis (*Barbar, 2008*; *Clark et al., 2018*; *Jurado et al., 2012*; *Rapali et al., 2011b*; *Singh et al., 2020*; *Slevin et al., 2014*; *Zaytseva et al., 2014*), prompting us to investigate a potential role of Ctp in transposon silencing.

Similarly to Panx, germ cell-specific shRNA-mediated knockdown of *ctp* resulted in significant de-repression of the germline-specific transposons *HeTA* and *burdock*, assayed by qPCR (*Figure 1C*). To assess global transcriptional changes upon Ctp loss, we performed RNA sequencing (RNA-seq) from ovaries from these knockdowns and compared transposon and gene expression to knockdown of *white* (*w*). We found that 49 out of 69 (71.0%) transposon families were significantly upregulated upon depletion of Ctp (*Figure 1D*) indicating that loss of Ctp impairs transposon repression in the germline. This effect was stronger than germline knockdown of Panx, which resulted in de-repression of 19.4% of transposon families (*Figure 1E*). Unlike Panx (63 or 0.5% genes), Ctp loss also had a strong effect on gene expression, with 2578 (or 16.2%) genes changing more than fourfold compared to the control. Piwi nuclear localisation was unaffected upon *ctp* germline knockdown (*Figure 1F* and *Figure 1—figure supplement 1B*), and piRNA levels were only moderately changed in ovarian somatic cells depleted of Ctp (*Figure 1—figure supplement 1C*), both consistent with the knockdown of a protein involved in TGS. However, germline depletion of Ctp resulted in a severe disruption of the ovary morphology with most ovarioles lacking late-stage egg chambers (*Figure 1G*), likely reflecting the requirement of this protein for a number of cellular processes (*Barbar, 2008*; *Rapali et al., 2011b*). Indeed, females with Ctp depleted in germ cells failed to lay eggs, while germline knockdown of *panx* resulted in sterility with only a minimal effect on egg laying (*Figure 1—figure supplement 1D*). Thus, the severe effect of *ctp* knockdown on transposon expression may also reflect pleiotropic effects caused by its loss.

Given the disruption of tissue architecture and widespread gene expression changes caused by in vivo depletion of Ctp, we turned to cultured ovarian somatic cells (OSCs) which harbour a functional Piwi-dependent nuclear transposon silencing pathway and provide a robust in vitro model for the somatic compartment of the ovary (*Saito et al., 2009*). We performed RNA-seq from cells depleted of Ctp via siRNA-mediated knockdown and compared gene and transposon expression to *siGFP* and *siPanx* controls (*Figure 2A,B*). Overall, 37 transposon families were strongly upregulated ($\log_2$ fold-change >2, RPKM >1, adjusted p value < 0.05) upon *ctp* knockdown, including both soma-specific transposons such as *gypsy* and *297*, and transposons active in both the germline and soma, such as *412* and *blood* (*Figure 2A,C,F*). This effect is considerably stronger when compared to knockdown of *panx*, *nxf2* or *piwi* (*Figure 2B,C*), which likely reflects the fact that Ctp is involved a number of essential cellular processes whereas Panx and Nxf2 have piRNA pathway-specific functions (*Batki et al., 2019*; *Fabry et al., 2019*). In line with this and consistent with the germline phenotype, knockdown of *ctp* had a greater impact on non-transposon gene expression, with 354 out of 15,644 (2.7%) genes changing significantly, compared to 0.7%, 0.6%, and 1.0% for *siPanx*, *siNxf2*, and *siPiwi*, respectively (*Figure 2D* and *Figure 2—figure supplement 1A*). Remarkably, in all four knockdowns up to half of the upregulated genes reside in genomic locations with their promoters

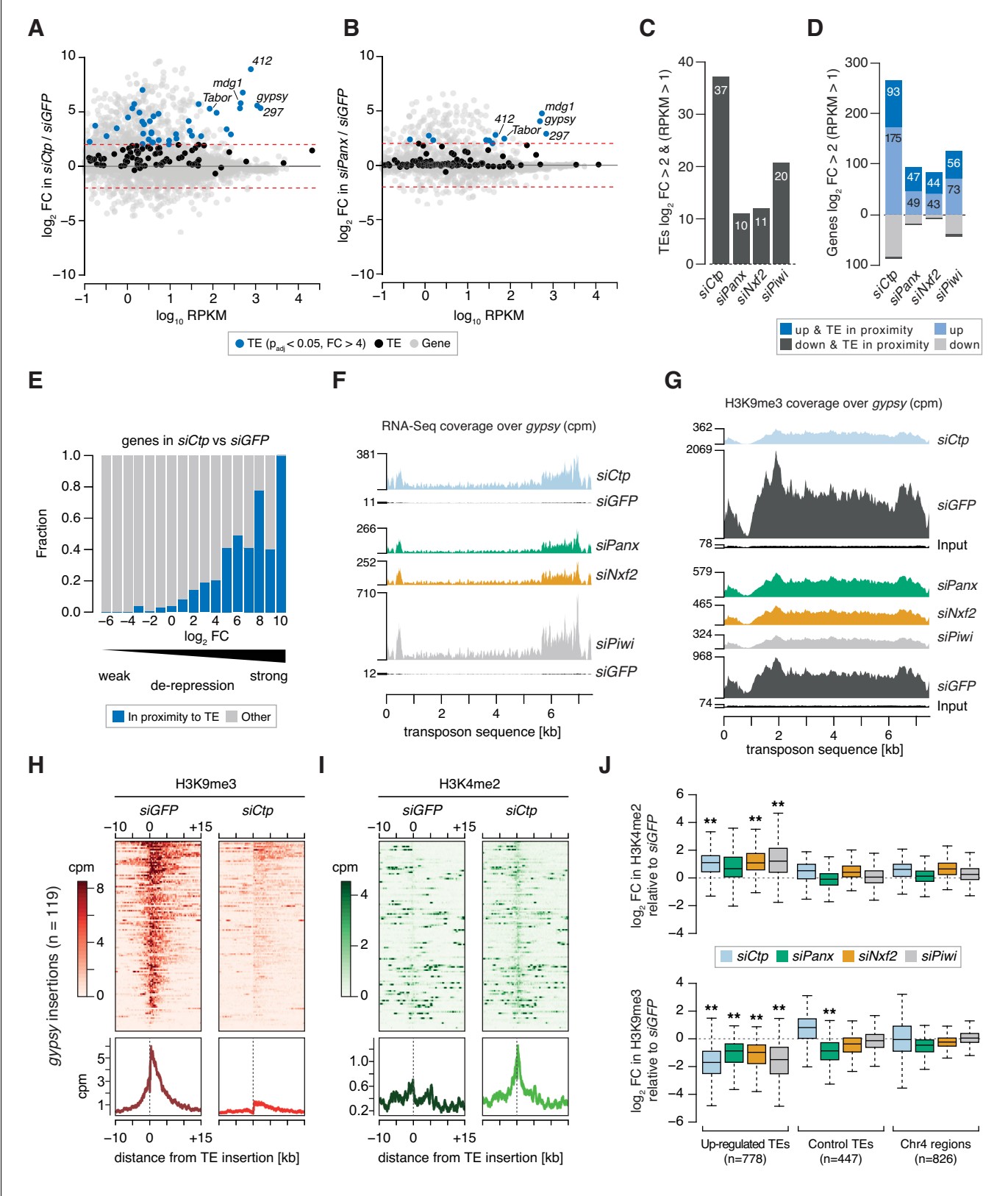

**Figure 2.** Ctp is required for transposon silencing and H3K9me3 deposition at transposon loci in ovarian somatic cells. (A) MA plot showing expression level against fold-change (FC) for genes and transposons in *siCtp* vs *siGFP*. Red dotted lines indicate log2FC = 2. TE, transposable element. (B) As in A but comparing *siPanx* vs *siGFP*. (C) Bar graph showing the number of transposable element (TE) families de-repressed more than fourfold in the

*Figure 2 continued on next page*

*Figure 2 continued*

indicated knockdowns (RPKM >1). (**D**) Bar graph showing the number of mis-expressed genes in the indicated knockdowns, highlighting those with their promoters in proximity to a *gypsy*, *mdg1*, *297*, *blood* or *412* insertion (RPKM >1). (**E**) Bar graph showing the proportion of genes with promoters in proximity to a *gypsy*, *mdg1*, *297*, *blood* or *412* insertion according to their fold-change in *siCtp* vs *siGFP* (RPKM >1). TE, transposable element. (**F**) Coverage plots showing normalised reads from RNA-seq over the *gypsy* transposon consensus sequence for the indicated OSC knockdowns. (**G**) As in F but showing coverage plots of H3K9me3 ChIP-seq reads for the indicated OSC knockdowns. Reads from input libraries are shown below. (**H**) Heatmaps (top) and meta-profiles (bottom) showing H3K9me3 ChIP-seq signal in the 25 kb surrounding 119 *gypsy* insertions in OSCs (sorted for decreasing signal in the *siGFP* control). (**I**) As in H for H3K4me2 signal. (**J**) Boxplot showing the fold-change in H3K9me3 and H3K4me2 signal (cpm) in the indicated knockdown compared to *siGFP* in H3K9me3 and H3K4me2 signal across 1 kb genomic bins surrounding insertion sites of upregulated transposons (TEs; *gypsy*, *mdg1*, *297*, *blood*, and *412*) in OSCs. ** indicates >2 fold difference in median and p<0.001 (Wilcoxon rank sum test). The online version of this article includes the following source data and figure supplement(s) for figure 2:

**Figure supplement 1.** Transposon de-repression upon loss of Ctp is associated with changes at the chromatin level.

**Figure supplement 1—source data 1.** Differential expression analysis of OSC RNA-seq data used for plots in *Figure 2* and *Figure 2—figure supplement 1*.

within close proximity of a *gypsy*, *mdg1*, *blood*, *297* or *412* insertion (*Figure 2D* and *Figure 2—figure supplement 1A*), suggesting that some of the observed gene expression changes are a pleiotropic effect caused by a failure in transposon silencing. Moreover, analysis of these genes in terms of fold-change revealed that the most highly upregulated genes are more likely to be in genomic proximity to a transposon than those that are changing weakly (*Figure 2E*), further indicating that piRNA pathway disruption is driving the strongest gene expression changes observed in these knockdowns. Thus, although knockdown of *ctp* results in expected effects on gene expression, we find that the largest expression changes in OSCs depleted of Ctp are related to transposons and therefore transposon control seems to be an essential role of Ctp in this cell type.

## H3K9me3 deposition at Piwi-regulated loci depends on Ctp

Given the high levels of transposon de-repression observed upon depletion of Ctp, we performed chromatin immunoprecipitation followed by sequencing (ChIP-seq) for H3K9me3 and H3K4me2 to assess whether the increased transcriptional output of transposons correlated with changes in the chromatin environment. Loss of Ctp resulted in a severe reduction of the H3K9me3 mark across derepressed transposon sequences including *gyspy* and *297*, in a similar manner to that observed upon *siPanx*, *siNxf2*, or *siPiwi*, which correlated with increased read coverage from RNA-seq (*Figure 2F,G* and *Figure 2—figure supplement 1B,C*). Analysis of the H3K9me3 signal surrounding *gypsy* and *297* insertions in the OSC genome revealed that Ctp depletion resulted in an almost complete loss of the H3K9me3 coverage from the genomic region surrounding each insertion (*Figure 2H* and *Figure 2—figure supplement 1D*). Moreover, this correlated with an increase in the coverage of H3K4me2, a mark indicative of active transcription, across the same insertions (*Figure 2I* and *Figure 2—figure supplement 1E*).

When considering ChIP-seq signal around all *gypsy*, *mdg1*, *blood*, *297*, or *412* insertions in the OSC genome, we detected a significant enrichment for H3K4me2 and depletion of H3K9me3 upon knockdown of PICTS complex components, while control regions (those surrounding unaffected transposons and heterochromatic regions on chromosome 4) were only marginally affected (*Figure 2J*). This effect is illustrated when looking at a *gypsy* insertion in the gene *expanded (ex)* on chromosome 2L (*Figure 2—figure supplement 1F*), which shows that increased transcriptional output across this locus correlates with loss of H3K9me3 and acquisition of H3K4me2 around the *gypsy* insertion site. Combined, these analyses show that Ctp contributes to the repressive chromatin states observed at transposon loci in OSCs, indicating that Ctp is required for TGS.

## Ctp is a bona fide co-transcriptional gene silencing effector

Our results implicate Ctp in co-transcriptional gene silencing of transposable elements (TEs) in the germline and somatic cells of the ovary. Consistent with a function in chromatin silencing, 3xFLAG-Ctp displayed both cytoplasmic and nuclear localisation in OSCs that was not perturbed upon knockdown of *panx* (*Figure 3A* and *Figure 3—figure supplement 1A*). Fractionation experiments from OSCs treated with *siGFP* confirmed the presence of Ctp in both nuclear and cytoplasmic compartments, with knockdown of *panx* not affecting the distribution or stability of endogenous Ctp

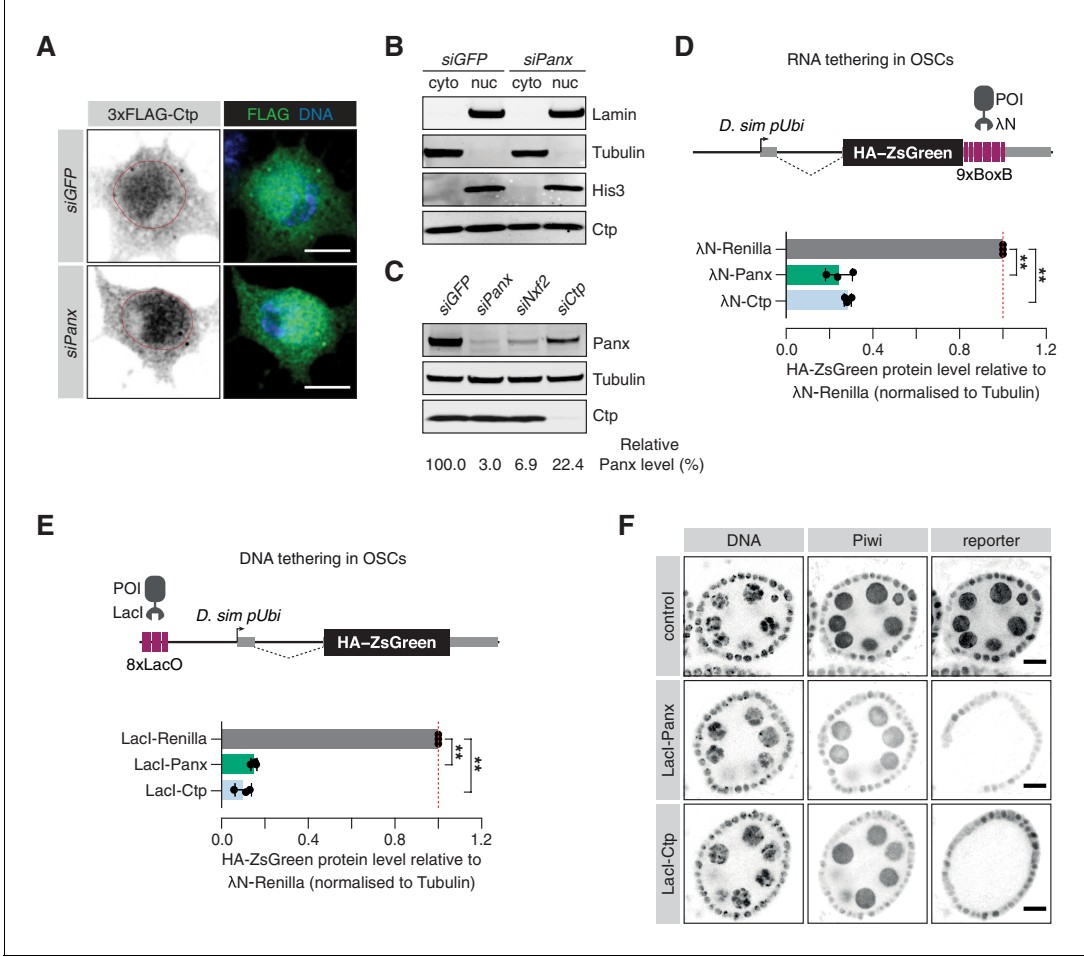

**Figure 3.** Ctp is involved in co-transcriptional gene silencing of transposons. (**A**) Immunofluorescence images showing the subcellular localisation of 3xFLAG-Ctp transiently transfected into OSCs upon the indicated knockdown. Scale bar = 5 μm. (**B**) Western blot for cytosolic proteins (Tubulin), nuclear proteins (Lamin and His3) and Ctp following subcellular fractionation of OSCs treated with the indicated siRNA. (**C**) Western blot showing Panx and Ctp protein levels in OSCs upon the indicated knockdown. The relative level of Panx protein, compared to *siGFP* and normalised to expression of Tubulin, is shown below. (**D**) Top: schematic showing the RNA tethering reporter system. Proteins of interest (POI) fused to the λN protein are recruited to the reporter mRNA (expressed from the *D. simulans* ubiquitin promoter) via BoxB sites in the 3'UTR. Bottom: bar graphs showing the level of HA-ZsGreen protein relative to the λN-Renilla control upon expression of λN-Panx or λN-Ctp. (**\*\***) p<0.001 (unpaired t-test). Error bars indicate standard deviation (n = 3). (**E**) Top: schematic showing the DNA tethering reporter system. Proteins of interest (POI) fused to the LacI DNA binding domain are recruited to the reporter locus via *LacO* sites upstream of the *D. simulans* ubiquitin promoter. Bottom: bar graphs showing the level of HA-ZsGreen protein relative to the LacI-Renilla control upon expression of LacI-Panx or LacI-Ctp. (**\*\***) p<0.001 (unpaired t-test). Error bars indicate standard deviation (n = 3). (**F**) Immunofluorescence images showing expression of Piwi compared to expression of the DNA tethering reporter in ovaries upon germline-specific expression of LacI-Panx and LacI-Ctp. Control indicates the parental stock expressing only the reporter. DNA is visualised with DAPI staining. Scale bar = 10 μm.

The online version of this article includes the following figure supplement(s) for figure 3:

**Figure supplement 1.** Ctp is a TGS factor.

(*Figure 3B*). However, depletion of Ctp resulted in a reduction of Panx protein levels, albeit not as severe as observed following knockdown of *nxf2* (*Figure 3C*), as reported previously (*Batki et al., 2019*; *Fabry et al., 2019*). Given this, we note that the transposon de-silencing effects observed upon knockdown of *ctp* in OSCs may in part be due to destabilisation of Panx.

A hallmark of proteins involved in co-transcriptional transposon silencing is their ability to induce transcriptional repression when artificially recruited to DNA encoding or nascent RNA transcribed from reporter constructs (*Sienski et al., 2015*; *Yu et al., 2015*). Many chromatin factors and transcriptional repressors exhibit silencing behaviour when tethered to DNA, including histone methyltransferases such as Egg and heterochromatin-associated proteins such as HP1a (*Batki et al., 2019*;

*Fabry et al., 2019*; *Sienski et al., 2015*). While DNA tethering systems provide a direct test of transcriptional silencing ability independently of nascent RNA, factors that affect co-transcriptional silencing in response to recognition of nascent RNA are also able to induce repression when artificially tethered to RNA, and Panx, Nxf2 and Nxt1 all elicit silencing activity in this context (*Batki et al., 2019*; *Fabry et al., 2019*; *Murano et al., 2019*; *Sienski et al., 2015*; *Yu et al., 2015*; *Zhao et al., 2019*).

To investigate a direct role of Ctp in TGS, we made use of two reporter systems in which proteins of interest are tethered to RNA (via BoxB hairpins) or to DNA (via LacO sites; *Figure 3D, E*; *Fabry et al., 2019*). The tethering reporters are both stably integrated into OSCs and the silencing ability of the protein of interest is assessed by evaluation of HA-ZsGreen protein and RNA levels after transfection of a λN- or LacI-fusion protein. Tethering of λN-Panx to RNA induced strong silencing of the reporter, both at the protein level (*Figure 3D* and *Figure 3—figure supplement 1B, C*) and the RNA level (*Figure 3—figure supplement 1D*) when compared to the λN-Renilla control, as reported (*Batki et al., 2019*; *Fabry et al., 2019*; *Murano et al., 2019*; *Sienski et al., 2015*; *Yu et al., 2015*; *Zhao et al., 2019*). Transfection of λN-Ctp also resulted in reporter repression at comparable levels to λN-Panx, with 28% ZsGreen protein and 29% nascent transcript signal remaining relative to the λN-Renilla control (*Figure 3D* and *Figure 3—figure supplement 1B–D*), indicating that Ctp can also function as a co-transcriptional repressor in OSCs. Moreover, tethering of Ctp to DNA resulted in potent reporter repression to a similar extent as tethering of Panx (*Figure 3E* and *Figure 3—figure supplement 1E,F*) with 10% ZsGreen protein and 4% ZsGreen transcript signal remaining relative to the control. In addition, use of an in vivo DNA tethering reporter showed that, like Panx, DNA recruitment of Ctp resulted in potent repression, indicating that Ctp can also function as a transcriptional repressor in the germ cell compartment of the fly ovary (*Figure 3F* and *Figure 3—figure supplement 1G*). Considered together, these data indicate that Ctp contributes to co-transcriptional repression of transposons.

## Two conserved motifs in Panx mediate the interaction with Ctp

Our results suggest that Ctp is a component of the PICTS complex and a bona fide TGS effector. In order to understand how Ctp contributes to Piwi-dependent transposon control, we performed immunoprecipitation of 3xFLAG-Ctp from OSCs and analysed recovered proteins by mass spectrometry. This confirmed the interaction between Ctp and Panx, Nxf2 and Nxt1 (*Figure 4A*, *Figure 4—figure supplement 1A,B* and *Figure 4—source data 1*). Of note, we did not find Piwi enriched (*Figure 4—figure supplement 1B*) consistent with results arising from immunoprecipitations of other members of PICTS (*Batki et al., 2019*; *Fabry et al., 2019*). Co-immunoprecipitation experiments in OSCs showed that the interaction between Ctp and Panx is mediated by the carboxy-terminal region of Panx (*Figure 4B*). A Panx mutant lacking the second coiled coil region (Panx$^{\Delta CC2+deg}$), which fails to interact with Nxf2 (*Fabry et al., 2019*), was still able to bind Ctp (*Figure 4—figure supplement 1C*), suggesting that the Panx-Ctp interaction is independent of Nxf2. In contrast, Ctp and Nxf2 were only found to associate in the presence of Panx (*Figure 4—figure supplement 1D*), pointing toward a direct interaction between Ctp and Panx, with Panx serving as an interaction platform for PICTS complex components.

In addition to PICTS components, we identified another 56 proteins that were significantly enriched (log$_2$FC > 2 and p<0.01) in the Ctp immunoprecipitation mass spectrometry (IP-MS). Of these, 20 have been reported previously (*Figure 4A,C*), including the dynein intermediate chain short wing (sw) and the transcription factor Asciz (*Clark et al., 2018*; *Jurado et al., 2012*; *Lo et al., 2001*; *Makokha et al., 2002*; *Nyarko et al., 2004*; *Zaytseva et al., 2014*). Analysis of Ctp partners across a range of species has shown that Ctp binds a linear recognition sequence found in intrinsically disordered protein regions (*Jespersen et al., 2019*; *Makokha et al., 2002*; *Rapali et al., 2011a*). This sequence, hereafter referred to as the TQT motif, comprises eight amino acids anchored by a highly conserved triplet, most commonly a glutamine residue sandwiched between two threonines (*Jespersen et al., 2019*; *Rapali et al., 2011a*). Among all 59 proteins enriched in the Ctp IP-MS, we identified 32 proteins with one or more putative TQT motifs (*Figure 4C,D* and *Supplementary file 1*) using the binding sequence prediction tool LC8Pred (*Jespersen et al., 2019*). Strikingly, this included two highly scoring and conserved recognition sequences in the C-terminus of Panx (*Figure 4E*).

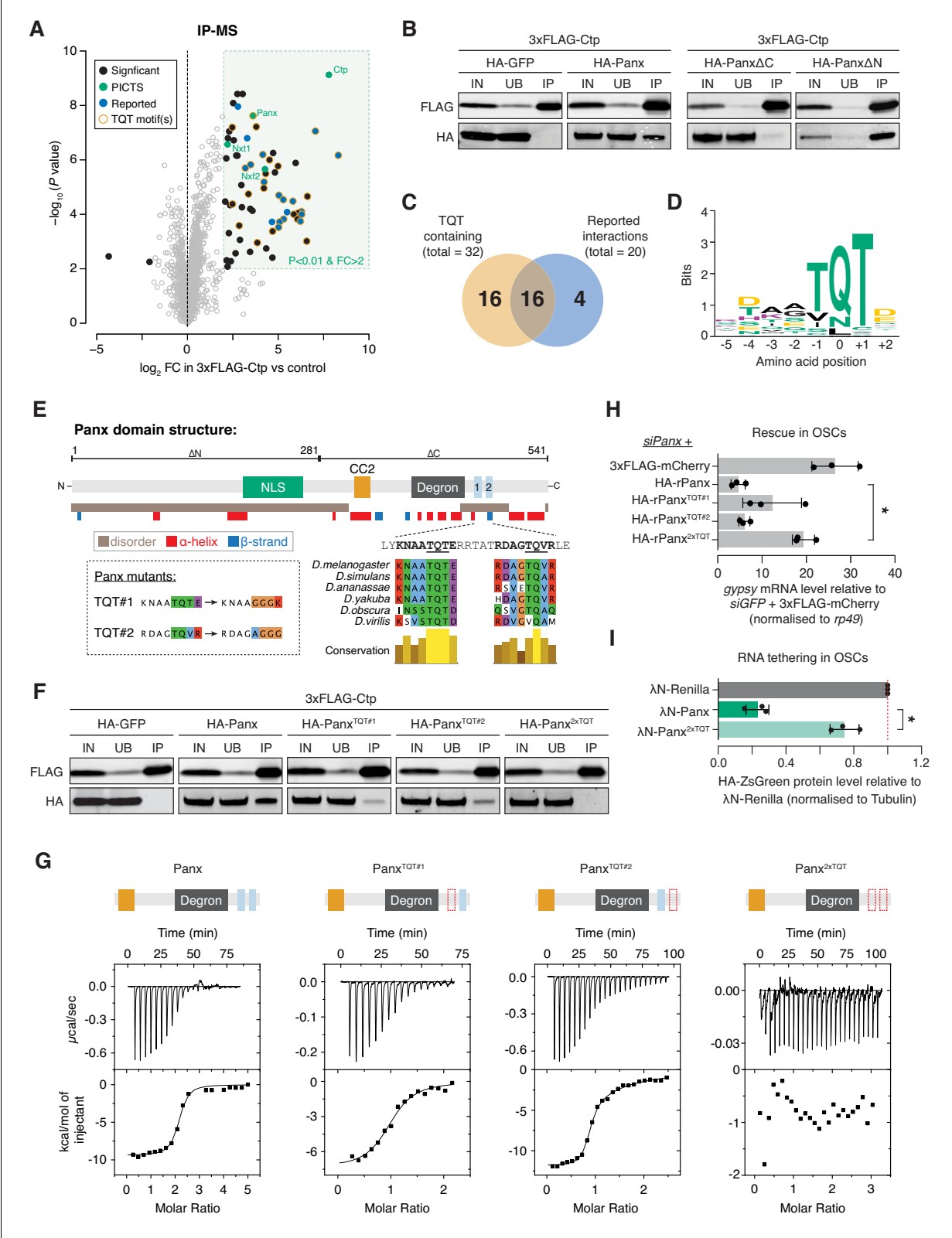

**Figure 4.** Ctp associates with PICTS via two highly conserved motifs in the carboxy-terminal region of Panx. (**A**) Volcano plot showing enrichment against significance for proteins identified by mass spectrometry that co-purify with 3xFLAG-Ctp from OSC lysates compared to the 3xFLAG-mCherry control (n = 3). (**B**) Western blot analysis for 3xFLAG-Ctp and indicated HA-tagged constructs following FLAG immunoprecipitation from OSCs. IN=input (1x), UB=unbound (1x), and IP=immunoprecipitate (10x). (**C**) Venn diagram for proteins significantly enriched in the Ctp IP-MS indicating the

*Figure 4 continued on next page*

*Figure 4 continued*

overlap between previously reported Ctp interactors and TQT motif-containing proteins. (D) Sequence logo representing the most common Ctp recognition motif found in proteins co-purifying with Ctp from OSCs. Letter height represents relative amino acid enrichment and letters are coloured according to amino acid property (positive charge=purple, negative charge=yellow, polar=green and hydrophobic=black). (E) Schematic showing the known functional domains of Panx, including the nuclear localisation signal (NLS), Nxf2 interacting region (CC2), degron and TQT motifs, alongside a disorder and secondary structure prediction. TQT motif conservation across *Drosophila* species and mutations made in these sequences are indicated in the inset. (F) As in B. IN=input (1x), UB=unbound (1x), and IP=immunoprecipitate (10x). (G) Isothermal titration calorimetry thermograms for Ctp with Panx, Panx$^{TQT#1}$, Panx$^{TQT#2}$, and Panx$^{2xTQT}$. A schematic showing the Panx fragment (domains as in E) used in each experiment is indicated above. The red dotted outline indicates mutation of the TQT motif. (H) Bar graph showing relative fold-changes in OSC mRNA levels of the soma-specific transposon *gypsy* upon knockdown of panx and re-expression of the indicated rescue construct. (*) p<0.05 (unpaired t-test). Error bars indicate standard deviation (n = 3). (I) Bar graphs showing the level of HA-ZsGreen protein relative to the λN-Renilla control upon tethering of λN-Panx or λN-Panx$^{2xTQT}$ to the RNA reporter in OSCs. (*) p<0.05 (unpaired t-test). Error bars indicate standard deviation (n = 3).

The online version of this article includes the following source data and figure supplement(s) for figure 4:

**Source data 1.** Source data for volcano plot shown in *Figure 4A*.
**Figure supplement 1.** Two recognition motifs in the Panx C-terminus interact with Ctp.
**Figure supplement 1—source data 1.** Source data for volcano plot in *Figure 4—figure supplement 1K*.

To test if either of the predicted TQT motifs in Panx are bound by Ctp, we generated Panx mutants in which either the first (Panx$^{TQT#1}$), the second (Panx$^{TQT#2}$), or both motifs (Panx$^{2xTQT}$) were disrupted by point mutations in the anchor triplet (*Figure 4E*). Panx$^{2xTQT}$ no longer contained any predicted Ctp-binding sequences when searched using LC8Pred. Co-immunoprecipitation analysis in OSCs revealed that both Panx$^{TQT#1}$ and Panx$^{TQT#2}$ have a reduced ability to interact with Ctp, but only mutation of both sites entirely abolishes the interaction (*Figure 4F*). Panx$^{2xTQT}$ was still able to interact with Nxf2, supporting that the interactions of Panx with Nxf2 and with Ctp are independent (*Figure 4—figure supplement 1E*).

To investigate whether both TQT sites within Panx are occupied by Ctp simultaneously, we performed isothermal titration calorimetry (ITC) to assay the stoichiometry and binding kinetics of the Panx-Ctp interaction in vitro. ITC for a Panx fragment encompassing the Nxf2 interaction region (CC2) (*Fabry et al., 2019*), the previously annotated degron (*Batki et al., 2019*) and both TQT motifs confirmed that both sites are bound by Ctp in vitro (*Figure 4G*, *Figure 4—figure supplement 1F* and *Supplementary file 2*). The wild-type Panx fragment bound Ctp at a Panx:Ctp stoichiometry of 1:2 and with high affinity (K$_D$ = 0.1 µM±0.01 µM) (*Figure 4G*). In line with the co-immunoprecipitation analysis, Ctp binding was completely abolished upon mutation of both TQT motifs (*Figure 4G*). ITC for the same fragment containing mutations in either of the TQT motifs individually showed that each site exhibited Ctp binding alone at a Panx:Ctp ratio of 1:1 (*Figure 4G*). Both sites bind Ctp strongly, although the first site is bound with higher affinity than the second (K$_D$ = 0.7 µM±0.10 µM and K$_D$ = 0.2 µM±0.06 µM upon mutation of TQT#1 and TQT#2, respectively). These data are in agreement with our co-immunoprecipitation analysis in OSCs which found that mutation of TQT#1 more strongly impacted the interaction of Panx and Ctp than mutation of TQT#2 (*Figure 4F*). Overall, this supports a model in which the TQT motifs are bound cooperatively and has been observed in other Ctp-binding partners with two recognition sites in close proximity (*Hall et al., 2009*; *Nyarko et al., 2013*).

## Panx depends on Ctp to support transposon silencing

Having identified point mutations in Panx that specifically disrupt Ctp binding, we utilised the Panx$^{2xTQT}$ mutant to investigate the PICTS complex-specific function of Ctp, bypassing the pleiotropic effects caused by *ctp* knockdown. Depletion of Panx in OSCs resulted in de-repression of *gypsy* that could be rescued by expression of siRNA-resistant wild-type Panx (*Figure 4H*). Panx$^{2xTQT}$ failed to rescue *gypsy* repression in cells depleted of endogenous Panx (*Figure 4H*), despite showing comparable expression levels (*Figure 4—figure supplement 1G*) and localising to the nucleus (*Figure 4—figure supplement 1H*), implying that Ctp is required for Panx to direct transposon silencing. As predicted from higher affinity binding of Ctp to TQT#1 (*Figure 4G*), mutating this site had a stronger impact on the ability of Panx to rescue transposon silencing than mutation of TQT#2 (*Figure 4H*). Interestingly, this correlates with greater conservation of TQT#1 across *Drosophila* species (*Figure 4E*). Consistent with an inability to repress transposons, Panx$^{2xTQT}$ also failed to induce

silencing of the RNA tethering reporter in OSCs (*Figure 4I* and *Figure 4—figure supplement 1I,J*), suggesting that interaction with Ctp is required for silencing upon enforced recruitment to target RNA. Lastly, comparing IP-MS for Panx with Panx$^{2xTQT}$ in OSCs showed that the effect of the mutation is highly specific. We did not find broad changes in the protein interactome of these constructs, with only Ctp significantly depleted in the Panx$^{2xTQT}$ immunoprecipitation (*Figure 4—figure supplement 1K* and *Figure 4—source data 1*). Our data indicate that Panx directly interacts with Ctp via two conserved motifs in the Panx C-terminus. In the absence of Ctp, Panx still associates with other members of PICTS and localises to the nucleus, but cannot support transposon repression, indicating that Ctp has a specific role within PICTS-directed TGS.

## Ctp induces silencing via Panx

*Drosophila* S2 cells do not express a functional piRNA pathway, although they do express several proteins implicated in TGS to varying degrees (*Figure 5A*). We reasoned that we could use S2 cells as an alternative context in which to probe the underlying mechanisms whereby selected components of PICTS interact to induce transcriptional repression. Using the DNA tethering reporter assay, we found that Panx, Nxf2, Ctp, and Egg, but not Piwi, were able to induce repression when recruited to the reporter locus in S2 cells (*Figure 5B* and *Figure 5—figure supplement 1A,B*). To confirm that the mechanism of transcriptional repression is faithfully recapitulated in this somatic cell type, we performed ChIP-seq for H3K9me3 and H3K4me2 in cells transfected with LacI-Panx. As a positive control for H3K9me3 deposition at the reporter locus we used the H3K9 methyltransferase Egg. Compared to the LacI-Renilla control, tethering of both LacI-Panx and LacI-Egg resulted in substantial accumulation of H3K9me3 across the reporter sequence and a concurrent reduction in H3K4me2 coverage (*Figure 5C*). This demonstrates that, as in OSCs, Panx-induced transcriptional silencing in S2 cells is correlated with formation of a repressive chromatin environment.

We noticed that the ability of Ctp to induce reporter silencing in S2 cells was considerably weaker than that of Panx, with ~40% signal remaining upon transfection of LacI-Ctp compared to ~10% for LacI-Panx (*Figure 5A* and *Figure 5—figure supplement 1C*). This was in contrast to the OSC context where both proteins induced silencing to similar extents (*Figure 3E*). We hypothesised that LacI-Ctp induces silencing via recruitment of Panx and that the ability of Ctp to silence the S2 cell reporter was therefore limited by the lower levels of endogenously expressed Panx in S2 cells compared to OSCs (*Figure 5A*). In support of this, overexpression of Panx but not the Panx$^{2xTQT}$ mutant with LacI-Ctp enhanced the level of reporter repression by fourfold, resulting in comparable silencing to that caused by LacI-Panx itself (*Figure 5D* and *Figure 5—figure supplement 1C,D*). Moreover, adjusting the plasmid transfection ratio to titrate the level of Panx overexpression showed that this effect was dosage-dependant, with lower levels of Panx expression correlating with a weaker silencing effect (*Figure 5D* and *Figure 5—figure supplement 1C,D*). To test whether Panx depends on the downstream recruitment of Ctp to induce epigenetic silencing, we tethered LacI-Panx$^{2xTQT}$ to DNA in S2 cells. Panx that is unable to interact with Ctp induced silencing to the same extent as wild-type Panx when recruited directly to a DNA locus (*Figure 5E* and *Figure 5—figure supplement 1E,F*), indicating that Panx does not absolutely require Ctp to elicit transcriptional repression in this setting. This is in line with previous reports showing that the Panx N-terminus, or Panx unable to interact with Nxf2, can alone induce silencing of DNA reporters (*Batki et al., 2019*; *Fabry et al., 2019*). Overall, our data supports a model in which Panx is the primary silencing effector within the PICTS complex that mediates recruitment, directly or indirectly, of chromatin modifiers.

## Ctp drives PICTS complex self-association

Ctp itself functions as a homodimer that binds two TQT motifs via two symmetrical binding grooves and as such facilitates the self-association of its interacting partners (*Barbar, 2008*; *Jespersen and Barbar, 2020*; *Rapali et al., 2011b*). We observed via co-immunoprecipitation that Panx self-associates in OSCs (*Figure 6A*) and we therefore asked whether Ctp mediates this interaction. In control cells, HA-Panx but not HA-GFP was recovered when 3xFLAG-Panx was immunoaffinity purified, however knockdown of *ctp* substantially reduced the co-immunoprecipitation of the differentially tagged Panx proteins (*Figure 6B*). Moreover, differentially tagged Panx$^{2xTQT}$ proteins failed to associate in OSCs (*Figure 6A*), whereas mutation of the first motif alone impaired, but was not sufficient to abolish the interaction (*Figure 6—figure supplement 1A*). Mutation of TQT#2 had a minimal effect on

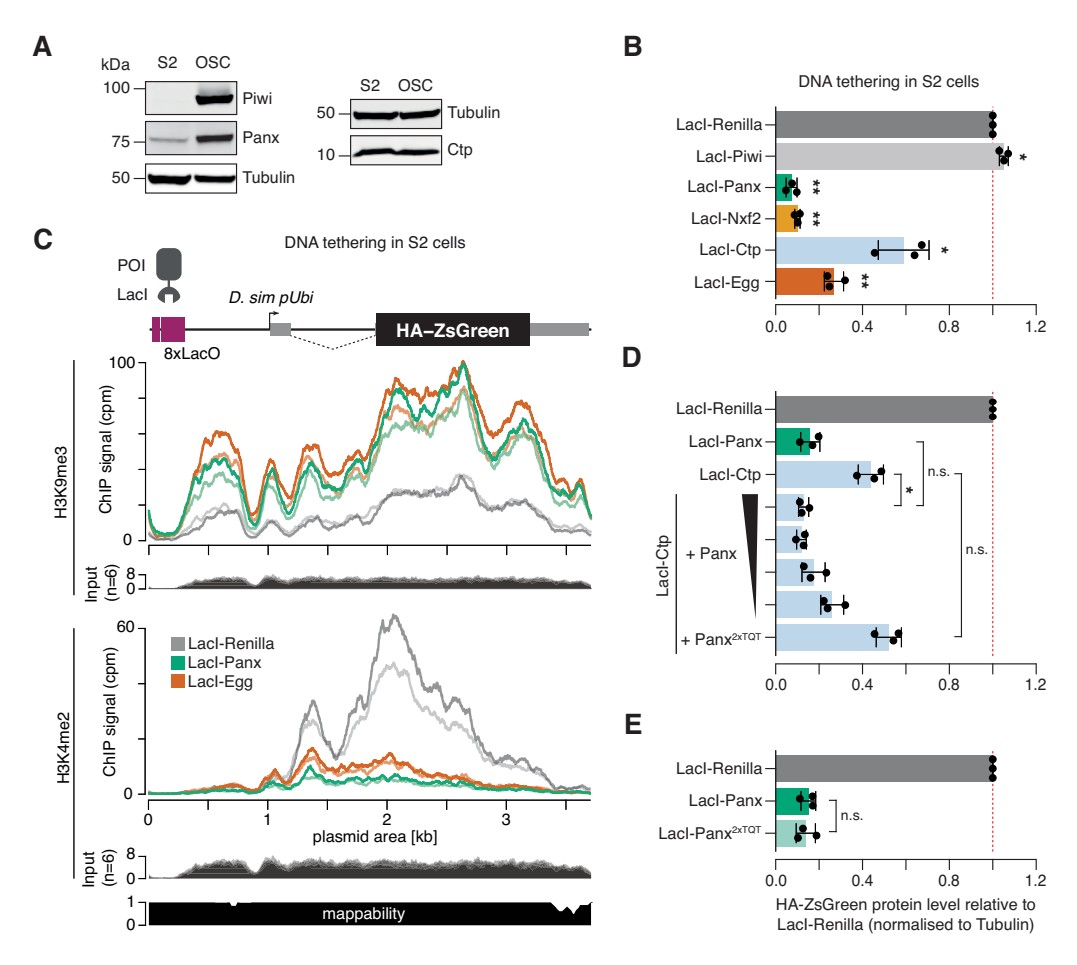

**Figure 5.** Ctp induces silencing via Panx. (**A**) Western blot analyses showing the relative level of Piwi, Panx, and Ctp in S2 cells compared to OSCs. Tubulin was used as a loading control. (**B**) Bar graphs showing the level of HA-ZsGreen protein relative to the LacI-Renilla control upon tethering of the indicated LacI-tagged protein to the DNA reporter in S2 cells. (*) $p<0.05$ (**) $p<0.001$ (unpaired t-test). Error bars indicate standard deviation (n = 3). (**C**) Coverage of H3K9me3 (middle) and H3K4me2 (bottom) ChIP-seq signal with corresponding input across the DNA tethering reporter locus. A schematic of the reporter is shown on the top and a mappability track is show below. (**D**) As in A with overexpression of Panx or Panx$^{2xTQT}$ indicated. The ratio of LacI-Ctp to HA-Panx plasmid ranges from 1:1 to 16:1 and is indicated by the black scale. (*) $p<0.05$ and n.s. = not significant (unpaired t-test). Error bars indicate standard deviation (n = 3). (**E**) Bar graphs showing the level of HA-ZsGreen protein relative to the LacI-Renilla control upon tethering of LacI-Panx or LacI-Panx$^{2xTQT}$ to the DNA reporter. n.s. = not significant (unpaired t-test). Error bars indicate standard deviation (n = 3).

The online version of this article includes the following figure supplement(s) for figure 5:

**Figure supplement 1.** Repression by Ctp is mediated via Panx.

the ability of Panx to dimerise in OSCs (*Figure 6—figure supplement 1A*), consistent with this mutant mostly rescuing transposon silencing upon endogenous Panx depletion (*Figure 4H*). Nxf2 also self-associated in OSCs in a manner dependent on its interaction with Panx, because it was disrupted by deletion of the UBA domain which mediates the Panx interaction (*Figure 6C*; *Batki et al., 2019*; *Fabry et al., 2019*).

We modelled the interaction of the Panx dual TQT peptide with Ctp based on known structures of Ctp homodimers bound to two intrinsically disordered peptides (*Figure 6D,E*; *Hall et al., 2009*; *Reardon et al., 2020*). In this model, each Ctp homodimer binds two Panx chains by binding exclusively to a single recognition site per Ctp in each chain resulting in a Panx:Ctp ratio of 2:4, consistent with our binding measurements by ITC. One Panx peptide is bound on each side of the dimerisation interface in a parallel manner and when bound by Ctp, each TQT motif adopts a β-strand conformation within a Ctp antiparallel β-sheet (*Figure 6D*). While only a model, the short linker between the TQT motifs could increase the steric hindrance between the bound Ctp dimers and result in minimal

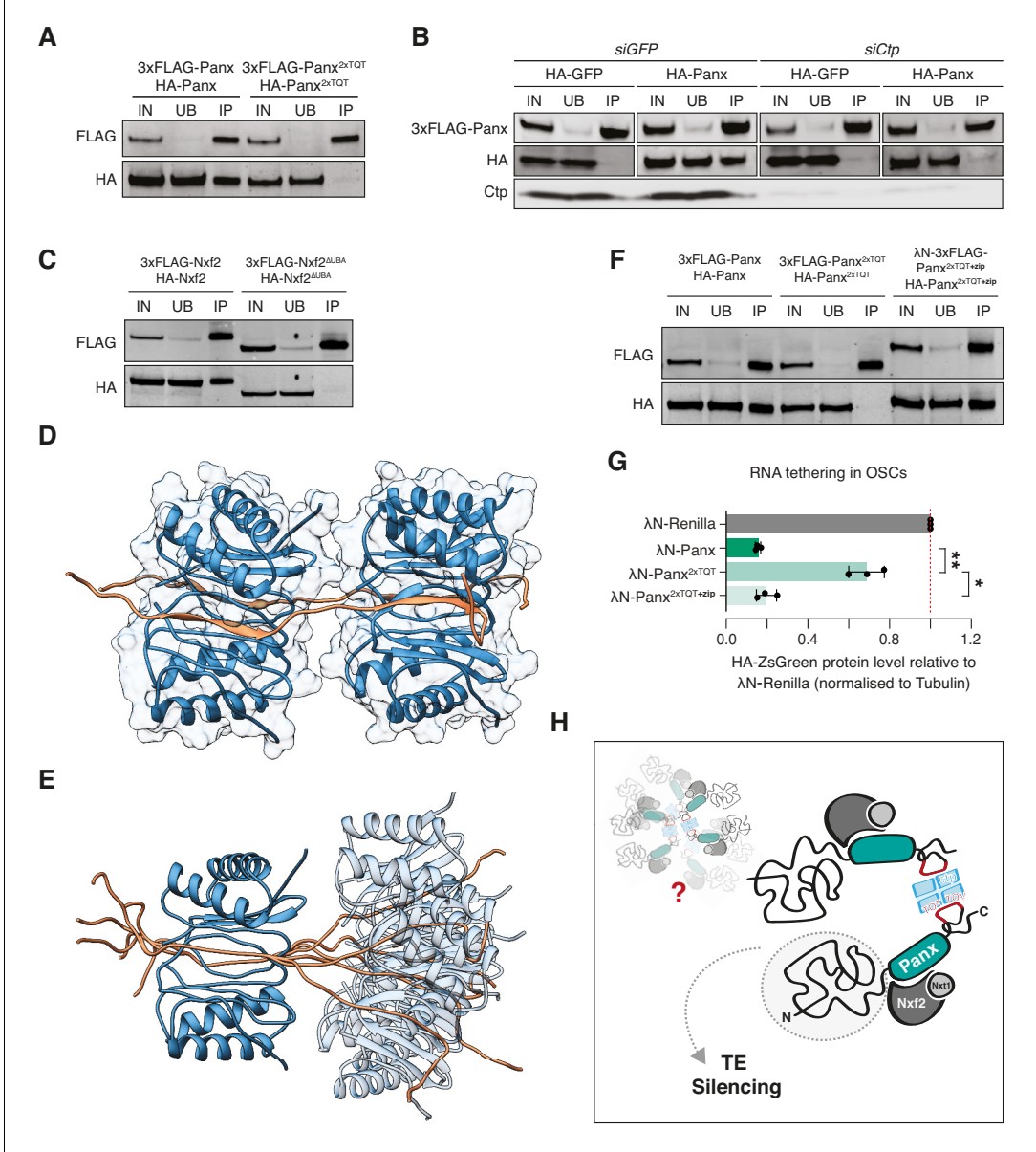

**Figure 6.** Ctp promotes higher order assembly of PICTS through dimerisation of Panx. (**A**) Western blot analysis for the indicated FLAG- and HA-tagged constructs following FLAG immunoprecipitation from OSCs. IN=input (1x), UB=unbound (1x), and IP=immunoprecipitate (10x). (**B**) Western blot analysis for 3xFLAG-Panx and either HA-GFP or HA-Panx following FLAG immunoprecipitation from OSCs treated with either *siGFP* or *siCtp*. IN=input (1x), UB=unbound (1x), and IP=immunoprecipitate (10x). (**C**) As in A. (**D**) and (**E**) Structural models showing two Panx peptides (amino acids 449–485) bound by two Ctp homodimers. For simplicity, a single structure from the ensemble is shown in D. The relative motional freedom of the Panx peptide when aligned on Ctp bound to the first TQT motif is shown in E. (**F**) As in A. (**G**) Bar graphs showing the level of HA-ZsGreen protein relative to the λN-Renilla control upon tethering of the indicated λN-Panx construct to the RNA reporter in OSCs. (*) p<0.05 (**) p<0.01 (unpaired t-test). Error bars indicate standard deviation (n = 3). (**H**) Model depicting possible PICTS complex organisation in which Ctp drives higher order assemblies of Panx-Nxf2-Nxt1 through dimerisation of the Panx C-terminus.

The online version of this article includes the following figure supplement(s) for figure 6:

**Figure supplement 1.** Dimerisation of Panx is essential for TGS.

flexibility between the two sites, thus potentially increasing the rigidity of the Panx peptide (*Figure 6E*). Overall, our combined evidence indicates that Ctp dimerises PICTS through its interaction with two binding motifs in the C-terminus of Panx. Whether the formation of a PICTS complex

dimer leads to the assembly of higher order oligomers as has been shown for other Ctp partners (*Slevin et al., 2014*) will require further investigation.

We reasoned that if the primary function of Ctp within PICTS is to promote complex dimerisation, and as such if the Panx[2xTQT] mutant failed to induce silencing when tethered to RNA due to a failure in self-association, then we could rescue this phenotype by enforcing dimerisation of Panx[2xTQT] via an alternative domain. To this end, we fused Panx[2xTQT] with a C-terminal leucine zipper from the *S. cerevisiae* transcription factor GCN4 (*Goldman et al., 2019*; *Sladewski et al., 2018*) which restored dimerisation of Panx[2xTQT] in OSCs (*Figure 6F*). Strikingly, tethering of this construct to RNA once again resulted in reporter repression (*Figure 6G* and *Figure 6—figure supplement 1B,C*), indicating that the main function of Ctp within PICTS is to promote dimerisation of the complex. Thus, our results indicate that self-assembly of the PICTS complex is a fundamental requirement for transcriptional silencing via nascent RNA association, and that this requirement has ultimately been fulfilled by the co-option of the dimerisation hub protein Ctp.

## Discussion

Recent work has revealed a recurring theme of co-option of existing gene expression machineries into the ovarian piRNA pathway (*Batki et al., 2019*; *ElMaghraby et al., 2019*; *Fabry et al., 2019*; *Kneuss et al., 2019*; *Murano et al., 2019*; *Zhang et al., 2012*; *Zhao et al., 2019*). The identification of Ctp as an essential component of the PICTS complex, here and by others (*Schnabl et al., 2021*), is another example and represents a key step in our understanding of piRNA-dependent co-transcriptional gene silencing. Studies across several species and of different macromolecular assemblies have uncovered the highly conserved function of Ctp as a dimerisation hub protein (*Jespersen and Barbar, 2020*), and high-affinity binding of Ctp to adjacent sites within Panx (*Figure 4F,G*) similarly promotes dimerisation of PICTS (*Figure 6A–C*). Thus, our data point to a model in which the carboxy-terminal domain of Panx acts as a scaffold for PICTS complex assembly through its interaction interfaces with Nxf2 (bridging to Nxt1) and Ctp, while the largely disordered amino-terminal region is the driving force for transcriptional silencing through mechanisms that remain unresolved (*Figure 6H*; *Batki et al., 2019*; *Fabry et al., 2019*).

Our results demonstrate a previously unknown requirement for dimerisation of the PICTS complex for it to accomplish transcriptional silencing, a functionality that has been enabled by the presence of Ctp-binding motifs within Panx. Dimerisation-incompetent Panx fails to induce silencing when artificially recruited to RNA (*Figure 4I*), while this mutant can trigger silencing to the same degree as wild-type Panx if tethered directly to DNA. It is, however, worth noting that when Panx is delivered to DNA via fusion to LacI, binding to the dyad symmetric *lacO* site effectively delivers the fusion protein as a tetramer (*Friedman et al., 1995*; *Lewis et al., 1996*). This is in contrast to the tethering of monomeric Panx as a λN-fusion to BoxB sites on RNA. Remarkably, artificial dimerisation of Panx[2xTQT] via fusion to a leucine zipper restored the ability of Panx to silence when tethered to RNA, consistent with the hypothesis that the function served by Ctp within PICTS is restricted to promoting assembly of a dimeric complex. Of note, Panx that is defective in Nxf2 binding also has a reduced ability to silence when tethered to RNA (*Batki et al., 2019*). Self-association is known to contribute to RNA binding in other Ctp-binding partners (*Goldman et al., 2019*). This might suggest that the interaction of an intact, dimeric complex with a nascent transcript is a requirement for silencing, potentially through preventing the dissociation of the transcript and the bound complex from chromatin. We speculate that anchoring of nascent transposon transcripts to chromatin ensures that silencing effectors have time to elicit heterochromatinisation specifically at the locus from which the transposon is transcribed.

To-date, all identified cofactors of Panx interact via well conserved regions in its C-terminus (*Figure 4B,E*; *Batki et al., 2019*; *Fabry et al., 2019*; *Murano et al., 2019*; *Zhao et al., 2019*). We have not yet identified proteins that interact with the poorly conserved, disordered N-terminus, which might suggest that mechanisms other than direct co-factor recruitment contribute to transcriptional repression. Notably, higher order oligomerisation of HP1a, for which dimerisation is a prerequisite, has been linked to the formation of phase separated droplets that mediate the formation of heterochromatin domains and heterochromatin-mediated gene silencing (*Larson et al., 2017*; *Strom et al., 2017*). Moreover, there are many characterised examples of liquid-liquid phase separation (LLPS) that are driven by RNA and their binding proteins, including those that are chromatin

associated (*Li and Fu, 2019*). Thus, we could envision the possibility that molecular links between dimeric PICTS and the nascent RNA might involve an LLPS-dependent silencing mechanism. We note that while this work was in revision, Brennecke and colleagues (*Schnabl et al., 2021*) reported that a recombinantly expressed, truncated version of PICTS/SFiNX can form molecular condensates in vitro in a Ctp-dependent manner. However, further investigation will be required to understand the contribution of the LLPS observed in vitro to transcriptional repression in vivo. Lastly, we found that artificial recruitment of Panx to DNA in somatic cells that lack a functional piRNA pathway is able to elicit heterochromatinisation of the reporter locus (*Figure 5B,C*), suggesting that the mechanisms and machineries operating downstream of Panx are not gonad-specific.

Ctp depletion had a stronger impact on transposon (and gene) expression than depletion of Panx in both germline and somatic tissue. We note that the involvement of Ctp in transposon regulation may not be restricted to its role within PICTS. Immunoprecipitation of Ctp from OSCs recovered a number of chromatin regulators including Wde and Egg. Wde contains a predicted Ctp recognition motif and therefore is likely a true partner of Ctp, highlighting the widespread presence of Ctp as a hub protein in a multitude of complexes. How does Ctp mediate the assembly of specific complexes without forming non-productive heterodimers between its different partners? Many known Ctp partners have additional self-association interfaces that are stabilised by Ctp binding and favour the formation of functional homodimers. For example, binding of Ctp to the dynein intermediate chain promotes self-association of a nascent helix domain (*Benison et al., 2006*), and a similar mechanism has been proposed for Panx (*Schnabl et al., 2021*). Lastly, the transcription factor Asciz which is known to regulate Ctp expression (*Jurado et al., 2012*; *Zaytseva et al., 2014*) was also enriched in the IP-MS and was previously linked to germline transposon silencing (*Czech et al., 2013*), a finding potentially explained by the implication of Ctp in this mechanism.

Ctp recognition sequences are well-conserved throughout putative insect Panx orthologs with some species such as the housefly, *Musca domestica*, and mosquito, *Aedes aegypti*, harbouring three predicted motifs compared to the two found in *Drosophilids* (*Figure 6—figure supplement 1D*). While for some Ctp partners, a single binding site is sufficient to promote their self-assembly, there are a number of examples where multiple motifs are bound cooperatively and contribute to increased structural organisation and rigidity of an otherwise disordered domain (*Clark et al., 2015*; *Reardon et al., 2020*). Thus, complex dimerisation could be a conserved requirement for transcriptional repression in insects. Clear Panx homologs or orthologs have not been identified outside of insects, though piRNA-dependent co-transcriptional silencing likely occurs throughout the animal kingdom. When functional equivalents of Panx are identified, it will be of great interest to determine whether the presence of essential TQT motifs indicates that engagement of Ctp as a dimerisation or multimerisation strategy is evolutionarily ancient or a more recent innovation in insects.

## Materials and methods

**Key resources table**

| Reagent type (species) or resource | Designation | Source or reference | Identifiers | Additional information |
|---|---|---|---|---|
| Gene (*Drosophila melanogaster*) | *ctp* | FlyBase | FBgn0011760 | |
| Gene (*Drosophila melanogaster*) | *panx* | FlyBase | FBgn0034617 | |
| Gene (*Drosophila melanogaster*) | *nxf2* | FlyBase | FBgn0036640 | |
| Gene (*Drosophila melanogaster*) | *nxt1* | FlyBase | FBgn0028411 | |
| Gene (*Drosophila melanogaster*) | *piwi* | FlyBase | FBgn0004872 | |
| Gene (*Drosophila melanogaster*) | *egg* | FlyBase | FBgn0086908 | |

*Continued on next page*

*Continued*

| Reagent type (species) or resource | Designation | Source or reference | Identifiers | Additional information |
|---|---|---|---|---|
| Cell line (*D. melanogaster*) | Ovarian somatic cells (OSC) | DOI:10.1038/nature08501 | RRID:CVCL_IY73 | |
| Cell line (*D. melanogaster*) | S2 cells | Thermo Fisher Scientific | Cat#: R69007 RRID:CVCL_Z232 | |
| Antibody | Anti-FLAG (Mouse monoclonal) magnetic beads | Sigma-Aldrich | Cat# M8823, RRID:AB_2637089 | |
| Antibody | Anti-GFP (Chicken polyclonal) | Abcam | Cat# ab13970, RRID:AB_300798 | IF(1:1,000) |
| Antibody | Anti-Piwi (Rabbit polyclonal) | DOI: 10.1016/j.cell.2007.01.043 | | IF(1:500) WB(1:2,500) |
| Antibody | Anti-HA (Mouse monoclonal) | Abcam | Cat# ab18181, RRID:AB_444303 | IF(1:1,000) |
| Antibody | Anti-FLAG (Rabbit monoclonal) | Cell Signaling Technology | Cat# 14793, RRID:AB_2572291 | IF(1:1,000) |
| Antibody | Anti-lamin (Mouse monoclonal) | Developmental Studies Hybridoma Bank | Cat# adl67.10, RRID:AB_528336 | IF(1:200) WB(1:200) |
| Antibody | Anti-FLAG (Mouse monoclonal) | Sigma-Aldrich | Cat# F1804, RRID:AB_262044 | WB(1:2,500) |
| Antibody | Anti-HA (Rabbit monoclonal) | Cell Signaling Technology | Cat# 3724, RRID:AB_1549585 | WB(1:2,500) |
| Antibody | Anti-tubulin (Rabbit polyclonal) | Abcam | Cat# ab18251, RRID:AB_2210057 | WB(1:2,500) |
| Antibody | Anti-tubulin (Mouse monoclonal) | Abcam | Cat# ab44928, RRID:AB_2241150 | WB(1:2,500) |
| Antibody | Anti-His3 (Mouse monoclonal) | Abcam | Cat# ab10799, RRID:AB_470239 | WB(1:1,000) |
| Antibody | Anti-DYNLL1 (Rabbit monoclonal) | Abcam | Cat# ab51603, RRID:AB_2093654 | WB(1:1,000) |
| Antibody | Anti-Panx (Mouse monoclonal) | DOI: 10.1101/gad.271908.115 | | WB(1:20) |
| Antibody | Anti-H3K9me3 (rabbit polyclonal) | Active Motif | Cat# 39161, RRID:AB_2532132 | |
| Antibody | Anti-H3K4me2 (rabbit polyclonal) | Merck Millipore | Cat# 07–030, RRID:AB_310342 | |
| Commercial assay or kit | TALON Metal Affinity Resin | Takara Bio | Cat# 635502 | |
| Commercial assay or kit | NEBNext Poly(A) mRNA magnetic Isolation Module | New England Biolabs | Cat# E7490L | |
| Commercial assay or kit | NEBNext Ultra Directional Library Prep Kit for Illumina | New England Biolabs | Cat# E7420L | |
| Commercial assay or kit | CleanTag Small RNA Library Preparation Kit | TriLink | Cat# L-3206 | |
| Commercial assay or kit | NEBNext Ultra II DNA Library Prep Kit for Illumina | New England Biolabs | Cat# E7645 | |
| Strain, strain background (*Escherichia coli*) | Rosetta (DE3) | Sigma-Aldrich | Cat# 70954 | |
| Recombinant DNA reagent | pET-24d(+) | Sigma-Aldrich | Cat# 69752 | |
| Recombinant DNA reagent | pVALIUM20 | DOI:10.1038/nmeth.1592 | | |

*Continued on next page*

*Continued*

| Reagent type (species) or resource | Designation | Source or reference | Identifiers | Additional information |
|---|---|---|---|---|
| Software, algorithm | Origin 7.0 | OriginLab | | |
| Software, algorithm | XPLOR-NIH | DOI:10.1002/pro.3248 | | https://nmr.cit.nih.gov/xplor-nih/ |
| Software, algorithm | Bowtie | DOI:10.1186/gb-2009-10-3-r25 | RRID:SCR_005476 | |
| Software, algorithm | STAR | DOI:10.1093/bioinformatics/bts635 | RRID:SCR_015899 | |
| Software, algorithm | deepTools | DOI:10.1093/nar/gkw257 | RRID:SCR_016366 | |
| Software, algorithm | DESeq2 | DOI:10.1186/s13059-014-0550-8 | RRID:SCR_015687 | |
| Software, algorithm | Trim Galore! | Felix Krueger | RRID:SCR_011847 | |
| Software, algorithm | LC8Pred | DOI:10.26508/lsa.201900366 | | http://lc8hub.cgrb.oregonstate.edu/LC8Pred.php |
| Software, algorithm | Image Studio Lite | LI-COR | RRID:SCR_013715 | |

## Cloning

All constructs were cloned using the NEBuilder HiFi DNA Assembly kit (New England Biolabs E2621) according to manufacturer's instructions. All HA-, 3xFLAG-, LacI-, and λN-fusion proteins were expressed in cells from the *Drosophila act5c* promoter. Genes were amplified from cDNA prepared from ovaries. A codon-optimised GCN4 leucine zipper sequence (AAL09032.1) was fused to the C-terminus of Panx$^{2xTQT}$ via PCR.

## Cell culture and transfection

OSCs (a gift from Mikiko Siomi) were cultured as described (*Niki et al., 2006*; *Saito, 2014*; *Saito et al., 2009*). S2 cells (Thermo Fisher Scientific; R69007) were cultured in Schneider's *Drosophila* medium (Gibco) supplemented with 10% Heat-Inactivated FBS (Sigma). Cell identity was authenticated in-house by high-throughput sequencing approaches and cell lines regularly tested negative for mycoplasma contamination in-house. Knockdowns in OSCs were performed as described previously (*Saito, 2014*). A list of siRNA sequences is provided in *Supplementary file 3*. Transfections in OSCs were carried out using either Xfect (Clontech) or the Cell Line Nucleofector kit V (Amaxa Biosystems) with the program T-029, as described (*Saito, 2014*). Rescue experiments in OSCs were carried out as described (*Fabry et al., 2019*; *Munafò et al., 2019*). S2 cells were transfected with the Cell Line Nucleofector kit V (Amaxa Biosystems) using the program G-030, as described (*Batki et al., 2019*). Stable cell lines were generated by co-transfection of the HA-ZsGreen reporter plasmid with a plasmid expressing a puromycin resistance gene under the control of the metallothionine promoter. Cell with reporter integrations were selected with puromycin treatment until stable lines were generated, with cell lines subsequently being cultured without puromycin.

## Fly stocks and handling

All flies were kept at 25°C on standard cornmeal or propionic food. Control $w^{1118}$ flies were a gift from the University of Cambridge Department of Genetics Fly Facility. For germline-specific knockdowns, we used a stock containing a UAS::Dcr2 transgene and a nos::GAL4 driver (described in *Czech et al., 2013*) and shRNA lines from the Bloomington *Drosophila* Stock Center, and crosses were done at 27°C. Fertility of females with germline knockdowns was scored by crossing six freshly hatched females to five $w^{1118}$ males and counting the number of eggs laid over a 3.5-day period and pupae that developed after 7 days. For the tethering experiments, we crossed stocks containing a *nanos*-driven LacI-fusion protein on chromosome two to a line expressing a GFP-Piwi in vivo tethering sensor on chromosome 3 (derived from JB313394; Vienna *Drosophila* Resource Center). Details on all fly stocks used in this study are listed in *Supplementary file 3*. A line expressing an shRNA against Panx (in pVALIUM20) was generated as described (*Ni et al., 2011*).

## Co-immunoprecipitation from OSC lysates

Four × 10$^6$ OSCs were nucleofected with HA- and 3xFLAG-tagged constructs (5 μg total) and harvested for coIP analysis 48 hr post-transfection. Cell pellets were lysed in 250 μl IP Lysis Buffer

(Pierce) supplemented with complete protease inhibitors (Roche) for 30 min at 4°C, and then spun at 21,100 g for 10 min at 4°C. Total protein was diluted to 1 ml with IP Lysis buffer and 5% was saved as input. The rest was incubated with 25 μl anti-FLAG M2 magnetic agarose beads (Sigma M8823) overnight at 4°C. Unbound fractions were collected, and the beads were washed in TBS 3 × 15 min at 4°C. Proteins were eluted from the beads in 20 μl 2X NuPAGE sample buffer (Invitrogen) by boiling at 90°C for 3 min prior to western blot analysis.

## RNA extraction and qPCR analysis

Cell pellets or dissected ovaries were lysed in 1 ml TRIzol Reagent (Invitrogen) and RNA extracted according to manufacturer's instructions. 1 μg of total RNA was treated with DNase I (Thermo Fisher Scientific) and reverse transcribed from oligo(dT)$_{20}$ primers (Integrated DNA Technologies) using the SuperScript III first strand synthesis kit (Thermo Fisher Scientific) according to manufacturer's instructions. Quantitative RT-PCR (qPCR) experiments were performed with Fast SYBR Green master mix (Applied Biosystems) according to manufacturer's instructions and analysed on a QuantStudio Real-Time PCR Light Cycler (Thermo Fisher Scientific). Transposon and ZsGreen reporter transcript levels were calculated using the ΔΔCT method (*Livak and Schmittgen, 2001*), normalised to *rp49* and fold-changes quantified relative to the indicated controls. All used qPCR primers are listed in *Supplementary file 3*.

## RNA-seq library preparation

Ten μg RNA was cleaned up using the RNeasy Miniprep Kit (Qiagen, 74106) according to manufacturer's instructions. One μg total RNA was used as input for library preparation as described (*Fabry et al., 2019*). Briefly, poly(A) mRNAs were isolated using the NEBNext Poly(A) mRNA magnetic Isolation Module (NEB, E7490L). Libraries were prepared using the NEBNext Ultra Directional Library Prep Kit for Illumina (NEB, E7420L) according to manufacturer's instructions and quantified with the KAPA Library Quantification Kit for Illumina (Kapa Biosystems) and sequenced on an Illumina HiSeq 4000.

## RNA-seq analysis

The RNA-seq data was sequenced as single-end 50 bp with four replicates per condition. For knockdown of *nxf2*, *panx*, and *piwi*, we reanalysed data from *Fabry et al., 2019* with three replicates per condition. Sequencing adapters (Illumina TruSeq) and low-quality bases were removed using Trim Galore! (v0.6.4) with '--stringency 6 -q 20'. Bowtie (v1.2.1.1) (*Langmead et al., 2009*) with '-S -n 2 -M 1 --best --strata --nomaqround --no-unal' was used to exclude reads matching known *Drosophila* viruses (https://obbard.bio.ed.ac.uk/data/Updated_Drosophila_Viruses.fas.gz, downloaded 13 January 2020), followed by alignment of the remaining reads (extracted using the additional --max and --un options) to a set of 122 transposon consensus sequences. All remaining reads were also aligned to the reference genome (*dm6*) using STAR (v2.7.3a) (*Dobin et al., 2013*) with '--outSAMtype BAM SortedByCoordinate --outMultimapperOrder Random --outSAMmultNmax 1 --outFilterMultimapNmax 1000 --winAnchorMultimapNmax 2000 --alignSJDBoverhangMin 1 --sjdbScore 3' and a genome index built using NCBI RefSeq.

To quantify gene and transposon expression, we used featureCounts with parameters '-s 2 -O --largestOverlap -Q 50' and gene annotations from Ensembl (release 97) or annotations corresponding to each transposon consensus sequence. This ensured that only reads mapping uniquely to the sense strand were counted. Differential expression analysis was done with the DESeq2 package (v1.26.0) (*Love et al., 2014*) from R Bioconductor. Gene expression analysis was performed using the standard DESeq2 workflow followed by log2 fold-change shrinkage using the 'ashr' method (*Stephens, 2017*). For analysis of transposon expression, we used the size factors from the gene expression analysis, whereas dispersion estimates and log2 fold-change shrinkage was calculated using a combined gene and transposon counts matrix to produce more robust estimates. RPKM values were calculated based on DESeq2's robust median ratio method in the 'fpm' function. A gene or transposon was considered differentially expressed if it had >4-fold difference in expression, adjusted p value < 0.05 and expression >1 RPKM. For a gene to be counted as near a transposon insertion, we required its TSS to be within 10 kb upstream and 15 kb downstream of the predicted insertion. Clustered insertion sites (less than 5 kb apart) were excluded from this analysis.

The bamCoverage module from deepTools (v3.3.2) (*Ramírez et al., 2016*) with '--binSize 1 --ignoreForNormalization chrM --normalizeUsing CPM --exactScaling --skipNonCoveredRegions --minMappingQuality 50' was used to create BigWig files for the UCSC Genome Browser tracks. A scaling factor was included to normalise to the total number of mapped reads. Coverage plots over the transposon consensus sequences were generated using the genomecov module from bedtools, counting all reads mapping to the sense strand of each transposon, followed by cpm normalisation and plotting using R. A list of all RNA-seq libraries generated and analysed in this study is given in *Supplementary file 4*.

## Small RNA-seq library preparation

To generate small RNA libraries, 18–29 nt small RNAs were sized selected by PAGE from 15 µg total RNA from OSCs. Following gel purification, libraries were prepared using the CleanTag Small RNA Library Preparation Kit (TriLink, L-3206) according to manufacturer's instructions. Libraries were quantified with the KAPA Library Quantification Kit for Illumina (Kapa Biosystems) and sequenced on an Illumina HiSeq 4000.

## Small RNA-seq analysis

The sRNA-seq data was sequenced as single-end 50 bp with two replicates per condition. Sequencing adapters (Illumina small RNA) and an abundant rRNA were removed by running Trim Galore! (v0.6.4), first with '--stringency 6 -q 0 --length 18', followed by '--stringency 30 -q 0 --length 18 -a TGCTTGGACTACATATGGTTGAGGGTTGTA'. Bowtie (v1.2.3) with '-S -n 2 -M 1 --best --strata --nomaqround --no-unal' was used to exclude reads mapping to *Drosophila* virus sequences, followed by alignment of the remaining reads (extracted using the additional --max and --un options) to a set of 122 transposon consensus sequences. Size distributions were calculated using all transposon-mapped reads and were similar across the two replicates. A list of all small RNA-seq libraries generated and analysed in this study is given in *Supplementary file 4*.

## Ovary immunostaining

Ovaries were dissected in ice-cold PBS, fixed in 4% paraformaldehyde (PFA; Alfa Aesar) for 14 min at room temperature, permeabilised with 3 × 10 min washes in PBS + 0.3% Triton (PBS-Tr) and blocked in blocking buffer (PBS-Tr + 1% BSA) for 2 hr at room temperature. Primary antibodies were diluted in blocking buffer and incubated overnight at 4℃. After 3 × 10 min washes at room temperature in PBS-Tr, secondary antibodies were diluted in blocking buffer and incubated overnight at 4℃. Samples were washed 4 × 10 min in PBS-Tr at room temperature with DAPI (1:5,000, Invitrogen D1306) added during the third wash. After 2 × 5 min washes in PBS, samples were mounted with ProLong Diamond antifade mountant (Thermo Fisher Scientific, P36961) and imaged on a Leica SP8 confocal microscope. The following antibodies were used: anti-GFP (1:1000; ab13970), anti-Piwi (1:500) (*Brennecke et al., 2007*). Secondary antibodies (1:500) were Alexa Fluor 488-, 555-, and 647-conjugated anti-mouse and anti-rabbit (Invitrogen) and Alexa Fluor 488-conjugated anti-chicken (Abcam).

## OSC immunostaining

Transfected OSCs were plated on fibronectin-coated coverslips overnight. Cells were washed with PBS, fixed in 4% PFA (Alfa Aesar) for 15 min at room temperature and rinsed in PBS. Fixed cells were permeabilised in PBS + 0.2% Triton X-100 for 10 min, rinsed with PBS and then incubated in PBS + 0.1% Tween-20 (PBST) + 1% BSA for 30 min. Primary antibodies were diluted in PBST + 0.2% BSA and incubated overnight at 4℃. Coverslips were washed 3 × 5 min in PBST and then incubated with secondary antibodies (Invitrogen) diluted in PBST + 0.2% BSA for 1 hr at room temperature. Coverslips were washed 3 × 5 min in PBST, stained with DAPI (1:1000 diluted in PBS, Invitrogen D1306) for 10 min, washed twice in PBS and mounted using ProLong Diamond Antifade Mountant (Thermo Fisher Scientific, P36961). The following antibodies were used: anti-HA (1:1000; abcam ab18181) anti-FLAG (1:1000; Cell Signaling Technology 14793S) and anti-Lamin (1:200; Developmental Studies Hybridoma Bank ADL67.10). Secondary antibodies (1:500) were Alexa Fluor 488-, 555- and 647-conjugated anti-mouse and anti-rabbit (Invitrogen).

## Tethering experiments and quantification

For RNA and DNA tethering experiments in OSCs, $4 \times 10^6$ cells with a stable integration of the reporter plasmid were nucleofected with 5 µg plasmid expressing the λN- or LacI- fusion protein as described above. After 48 hr, cells were pulsed a second time with 5 µg of the same plasmid and allowed to grow for an additional 48 hr before harvested for western blot and qPCR analysis. For tethering experiments in S2 cells, $4 \times 10^6$ cells with a stable integration of the DNA tethering reporter construct were nucleofected with 2 µg of LacI-fusion expression plasmid. After 72 hr, cells were harvested for western blot and qPCR analysis.

## Western blot

Cell pellets were lysed in RIPA buffer (Pierce) supplemented with protease inhibitors (Roche) and incubated for 20 min at 4℃. After centrifugation at 21,000 g for 10 min at 4℃, protein concentration was quantified using a Direct Detect Infrared Spectrometer (Merck). Ten to 20 µg total protein was separated on a NuPAGE 4–12% Bis-Tris denaturing gel (Thermo Fisher Scientific) and transferred to a nitrocellulose membrane using an iBLot2 dry transfer (Invitrogen). Primary antibody incubations were performed overnight at 4℃ and IRDye 680RD- and 800CW-conjugated secondary antibodies (LI-COR) were incubated for 45 min at room temperature. Primary antibodies used are as follows: anti-FLAG (1:2500; Sigma F1804), anti-HA (1:2500; Cell Signaling Technology C29F4), anti-Tubulin (1:2500; abcam ab18251), anti-Tubulin (1:2500; abcam ab44928), anti-Lamin (1:200; Developmental Studies Hybridoma Bank ADL67.10), anti-His3 (1:1000; abcam ab10799), anti-DYNLL1 (1:1000; abcam ab51603), anti-Piwi (1:2500) (*Brennecke et al., 2007*), and anti-Panx antibody (1:20) (*Sienski et al., 2015*). Images were acquired using an Odyssey CLx scanner (LI-COR) and processed/quantified in Image Studio Lite (LI-COR).

## Subcellular fractionation

The subcellular fractionation protocol was adapted from Batki and colleagues (*Batki et al., 2019*). Cell pellets resuspended in LB1 (10 mM Tris-HCl pH 7.5, 2 mM $MgCl_2$, 3 mM $CaCl_2$ supplemented with protease inhibitors [Roche]) for 5 min at 4℃ followed by centrifugation (1000 g, 5 min, 4℃). The supernatant was saved (swell fraction) and the cell pellet was resuspended and incubated in LB2 (10 mM Tris-HCl pH 7.5, 2 mM $MgCl_2$, 3 mM $CaCl_2$0.5% Nonidet P-40% and 10% glycerol, supplemented with protease inhibitors (Roche)) for 5 min at 4℃ followed by another centrifugation step (1000 g, 5 min, 4℃). The supernatant was saved (cytoplasmic fraction) and the nuclei pellet was resuspended in LB3 (50 mM Tris-HCl pH 8, 150 mM NaCl, 2 mM $MgCl_2$, 0.5% Triton X-100, 0.25% Nonidet P-40% and 10% glycerol, supplemented with protease inhibitors [Roche]). After a 10-min incubation on ice, the nuclear lysate was sonicated for 3 cycles of 30 s ON/30 s OFF using a Bioruptor Pico (Diagenode) to ensure solubilisation of chromatin and then spun 18,000 g for 10 min. The swell and cytoplasmic fraction was combined and the fractions were analysed by western blotting.

## Immunoprecipitation and mass spectrometry (IP-MS)

For 3xFLAG-Ctp IP-MS, $1 \times 10^7$ OSCs were seeded in 10 cm dishes, and transfected with 20 µg 3xFLAG- fusion expression plasmid (Xfect) the following day. Cells were harvested 48 hr post-transfection and lysed in IP Lysis Buffer (Pierce) supplemented with protease inhibitors (Roche) for 45 min at 4℃. After centrifugation at 16,500 g for 10 min at 4℃, lysates (with 5% saved as input) were incubated with 50 µl anti-FLAG M2 magnetic agarose beads (Sigma M8823) overnight at 4℃. Beads were washed three times for 15 min in IP Lysis Buffer, twice in ice-cold PBS and twice in 100 mM ammonium bicarbonate before being submitted for mass spectrometry analysis. For the 3xFLAG-Panx IP-MS, $1 \times 10^7$ OSCs were nucleofected with 5 µg 3xFLAG- fusion expression plus 2 µl siRNA against endogenous Panx. Cells were harvested 48 hr post transfection and fractionated according to the above protocol. After sonication, the nuclear fraction was additionally treated for 30 min with Benzonase (50 U/ml) at 4℃ followed by centrifugation. Nuclear lysate was incubated with 50 µl anti-FLAG M2 magnetic agarose beads (Sigma M8823) overnight at 4℃, washed three times for 15 min in LB3, twice in ice-cold TBS (pH 8.0) and twice in 100 mM ammonium bicarbonate before being submitted for mass spectrometry analysis.

## Mass spectrometry analysis

Samples were digested with trypsin and analysed on a Q-Exactive HF mass spectrometer (Thermo Fisher Scientific). Spectral. raw files were processed with the SequestHT search engine on Thermo ScientificTM Proteome Discoverer 2.2. Data was searched against a custom database derived from FlyBase ('dmel-all-translation-r6.24') at a 1% spectrum level FDR criteria using Percolator (University of Washington). The node for SequestHT included the following parameters: Precursor Mass Tolerance 20 ppm, Fragment Mass Tolerance 0.02 Da, Dynamic Modifications were methionine oxidation (+15.995 Da), asparagine and glutamine deamination (+0.984 Da). The Precursor Ion Quantifier node (Minora Feature Detector) included a Minimum Trace Length of 5, Max. ΔRT of Isotope Pattern 0.2 min. The consensus workflow included peptide validator, protein filter and scorer. For calculation of Precursor ion intensities, Feature mapper was set True for RT alignment, with the mass tolerance of 10ppm. Precursor abundance was quantified based on intensity and the level of confidence for peptide identifications was estimated using the Percolator node with a Strict FDR at q-value <0.01. Analysis of label-free quantification protein intensity data was carried out in R (version 3.6.1) using the qPLEXanalyzer package (version 1.2.0) (*Papachristou et al., 2018*). Only unique peptides identified with high confidence (peptide FDR < 1%) and mapping to a single protein were used for analysis. Peptide intensities were normalised between samples by median scaling within sample groups. For the Ctp IP-MS, peptides for which more than one target protein sample lacked measurements or for which more than three measurements were missing overall were discarded. For peptides with only one missing value across control samples, missing values were imputed using the nearest neighbour averaging (knn) imputation method provided in the R package MSnbase (version 2.10.1) (*Gatto and Lilley, 2012*). For peptides with more than one missing value across control samples, missing values were imputed using the deterministic minimum approach provided in MSnbase (version 2.10.1). For the Panx IP-MS, peptides for which all control samples lacked measurements or for which more than one target protein sample lacked measurements were discarded. Remaining missing values were then imputed using the nearest neighbour averaging (knn) imputation method provided in the R package MSnbase (version 2.10.1) (*Gatto and Lilley, 2012*). Peptide data was summarised to protein level by summing intensities for all peptides. Differential analysis was then carried out by linear modelling using the limma-based methods provided by the qPLEXanalyzer package. Multiple testing correction of p-values was applied using the Benjamini and Yekutieli method to control FDR (*Benjamini et al., 2001*).

## TQT identification and sequence logo generation

Significantly enriched proteins ($\log_2$ FC >2, p<0.01) were searched in LC8Pred (*Jespersen et al., 2019*) to identify putative TQT motifs (LC8Pred amino acid score >12). The final list of sequences (eight amino acids long) was used as input for sequence logo generation using https://weblogo.berkeley.edu/logo.cgi with default parameters.

## Sequence alignments and structural predictions

The domain structure of Panx was annotated according to previous studies (*Batki et al., 2019*; *Fabry et al., 2019*). A list of ortholog insect Panx species was adapted from Batki and colleagues (*Batki et al., 2019*). Sequences were aligned using Mafft (with default parameters) and visualised in Jalview (v.2.11.1.2). Secondary structural elements were predicted with SPIDER3 (*Heffernan et al., 2017*) and disorder predicted with NetSurfP2 (*Klausen et al., 2019*) accessed via quick2d (*Zimmermann et al., 2018*). TQT motifs were predicted in LC8Pred (*Jespersen et al., 2019*). The structural models were generated by fixing the interaction motif and Ctp coordinates to those observed in Protein Data Bank file 2P2T (*Benison et al., 2007*) and performing molecular dynamics in XPLOR-NIH (*Schwieters et al., 2018*).

## Protein expression and purification

Panx fragments for ITC were cloned into pET-24d(+) (Sigma 69752) with an N-terminal His6 tag and expressed in *E. coli* strain Rosetta(DE3) (Sigma 70954). Bacterial cultures for proteins expression were grown in lysogeny broth (LB) at 37°C to an optical density of 0.6 prior to induction with 1 mM of isopropyl-β-D-1-thiogalactopyranoside (IPTG). Protein expression proceeded for 6 hr at 18°C and cells were then harvested from the cultures by centrifugation. Cells were lysed by sonication and

then centrifuged a second time to clarify the lysate. The resulting supernatant was purified by immobilised metal affinity chromatography using TALON Metal Affinity Resin (Takara Bio USA, Mountain View, California). All purified protein samples were stored at 4°C with a protease inhibitor mixture of pepstatin A and phenylmethanesulfonyl fluoride and used within a week. SDS-PAGE gels run prior to the use of the samples showed no evidence of proteolysis.

## Isothermal titration calorimetry

Binding thermodynamics of the Panx and Ctp interaction were obtained at 25°C with a VP-ITC microcalorimeter (Microcal, Westborough, MA). The binding buffer was composed of 50 mM sodium phosphate, 150 mM sodium chloride, 5 mM β-mercaptoethanol, pH 7.5. Panx constructs were placed in the reaction cell at a concentration of 10–20 μM and titrated with Ctp at a concentration of 150–200 μM. Data for all were fit to a simple single-site binding model in Origin 7.0, allowing for determination of the stoichiometry ($N$), dissociation constant ($K_d$), and the change in enthalpy ($\Delta H$), and entropy ($\Delta S$). Protein concentrations were determined by absorbance measurements at 280 and 205 nm. Molar extinction coefficients for each construct at 280 nm were calculated with the ProtParam tool on the ExPASy website and are as follows *Gasteiger et al., 2005*. Ctp = 17,420 $M^{-1}cm^{-1}$, Panx, Panx$^{TQT\#1}$, Panx$^{TQT\#2}$ = 15,930 $M^{-1}cm^{-1}$. Similarly, molar extinction coefficients for measurements at 205 nm were computed with a Protein Calculator tool (*Anthis and Clore, 2013*).

## ChIP-seq library preparation

The ChIP-seq protocol was adapted from Schmidt and colleagues (*Schmidt et al., 2009*) and carried out from OSCs according to Fabry and colleagues (*Fabry et al., 2019*) using commercially available rabbit polyclonal anti-H3K9me3 (Active Motif, cat # 39161) and rabbit polyclonal anti-H3K4me2 (Merck Millipore, cat # 07–030) antibodies. ChIP from S2 cells was carried out using the same protocol from $2 \times 10^7$ cells with a modification to the chromatin preparation procedure to sonicate chromatin for 8 cycles of 30 s on/30 s off using a Bioruptor Pico (Diagenode). Recovered DNA was quantified on an Agilent TapeStation System using a High Sensitivity DNA ScreenTape (Agilent). DNA sequencing libraries were prepared with NEBNext Ultra II DNA Library Prep Kit for Illumina (NEB; E7645) according to the manufacturer's instructions and quantified with KAPA Library Quantification Kit for Illumina (Kapa Biosystems). OSC and S2 cell ChIP-seq libraries were sequenced on an Illumina HiSeq 4000 and NovaSeq 6000, respectively.

## ChIP-seq analysis

The ChIP-seq data from OSCs was sequenced as single-end 50 bp with two replicates per condition. For knockdowns of *nxf2*, *panx* and *piwi* we re-analysed data from *Fabry et al., 2019* with two replicates per condition. Sequencing adapters (Illumina TruSeq) and low-quality bases were removed using Trim Galore! (v0.6.4; using cutadapt v1.18) with '--stringency 6 -q 20'. Bowtie (v1.2.1.1) (*Langmead et al., 2009*) with '-S -n 2 -M 1 --best --strata --nomaqround --no-unal' was used to align the reads both to the reference genome (*dm6*) and to a set of 122 transposon consensus sequences. PCR duplicates were removed from the genomic alignment using MarkDuplicates from Picard tools (v2.21.2).

The bamCoverage module from deepTools (v3.3.2) (*Ramírez et al., 2016*) with '--binSize 1 --ignoreForNormalization chrM --normalizeUsing CPM --extendReads 260 --centerReads --exactScaling --minMappingQuality 255' was used to create BigWig files for the UCSC Genome Browser tracks and profile plots, which also shifted the reads to the expected midpoint of the insert. Data matrices for the heatmaps and profile plots around transposon insertions were constructed using the computeMatrix module in 'reference-point' mode with '--missingDataAsZero -b 10000 -a 15000 --referencePoint center' and using bin size five for profiles and 50 for heatmaps. Plotting was done in R and 95% confidence intervals of the mean were calculated using bootstrapping with 200 bootstrap samples. For readability, heatmap values were capped at the 99th percentile. Coverage plots over the transposon consensus sequences were generated using the genomecov module from bedtools, counting all reads mapping to each transposon, followed by cpm normalisation and plotting using R.

For the binning analysis, the genome was divided in 1 kb non-overlapping bins, only keeping bins with mappability >50%. Bins with the following features were identified; (i) near insertion (within 1 kb) of up-regulated transposons (*gypsy*, *mdg1*, *blood*, *297*, *412*), (ii) near insertion of other

transposons (*Juan*, *Bari1*, *1360*, *pogo*, *Burdock*, *diver*), or (iii) in the heterochromatin region on chromosome four and more than 10 kb away from any insertion. For each knockdown, reads mapping to each bin were counted, normalised for mappability and library size differences, and averaged across the two replicates. Each knockdown was compared to its respective control using Wilcoxon rank sum test.

The reporter assay in S2 cells was sequenced as paired-end 2 × 50 bp with two replicates per condition. Trimming and alignment was done as described above, except that Bowtie (v1.2.3) with the additional options '-y --maxins 500 --fr --allow-contain' were used and that alignment was done against the plasmid sequence concatenated to the *dm6* genome. Duplicates were removed using MarkDuplicates and read coverage over the reporter was extracted using bamCoverage with '--binSize 1 --ignoreForNormalization chrM --normalizeUsing CPM --extendReads --centerReads --exactScaling --outFileFormat bedgraph --region pBC470:1:6355'. A list of all ChIP-seq libraries generated and analysed in this study is given in *Supplementary file 4*.

### Mappability tracks

All mappability tracks were constructed to reflect the read length and alignment strategy used in each analysis. In short, we used 'bedtools makewindows -w *n* -s 1 -g genome.fa.fai > windows.bed' followed by 'bedtools getfasta -fi genome.fa -bed windows.bed' to construct all possible *n*-mers of the reference genome, where *n* was the expected read length. For most libraries, *n* was equal to 50. For the paired-end 2 × 50 bp data, we approximated mappability both using *n* = 100 and by constructing all possible pairs of 50-mers separated by the mean distance between read pairs. The second mates were created using 'bedtools shift' with '-s 146' to move the coordinates, followed by extraction of the reverse complement sequence. The genome-derived *n*-mers or pairs or *n*-mers were aligned to the reference genome with the software and options used to align the corresponding sequencing data. Mappability was calculated as the number of uniquely mapped reads covering a coordinate, divided by *n*. For the paired-end 2 × 50 bp reads, mappability was estimated as the mean mappability from the two approaches described above.

## Acknowledgements

We thank members of the Hannon group for discussion. We thank Martin H Fabry for the list of OSC transposon insertions and Ashley Sawle for contributions to analysis of the proteomics data. We thank the CRUK Cambridge Institute Bioinformatics, Genomics, Microscopy, RICS and Proteomics Core Facilities for technical support. We thank the University of Cambridge Department of Genetics Fly Facility for microinjection services and fly stock generation. We thank the Vienna *Drosophila* Resource Center and the Bloomington Stock Center for fly stocks. We thank Mikiko Siomi for OSCs and Julius Brennecke for anti-Panx antibody. Research in the Barbar laboratory is supported by the National Science Foundation (Award 1617019). MM was supported by a Boehringer Ingelheim Fonds PhD fellowship. GJH is a Royal Society Wolfson Research Professor (RP130039). Research in the Hannon laboratory is supported by Cancer Research UK (A21143) and a Wellcome Trust Investigator award (110161/Z/15/Z).

## Additional information

### Funding

| Funder | Grant reference number | Author |
|---|---|---|
| Cancer Research UK | Core funding (A21143) | Gregory J Hannon |
| Wellcome Trust | Investigator award (110161/Z/15/Z) | Gregory J Hannon |
| National Science Foundation | Award 1617019 | Elisar J Barbar |
| Royal Society | Wolfson Research Professor (RP130039) | Gregory J Hannon |
| Boehringer Ingelheim Fonds | PhD fellowship | Marzia Munafò |

The funders had no role in study design, data collection and interpretation, or the decision to submit the work for publication.

## Author contributions
Evelyn L Eastwood, Conceptualization, Data curation, Formal analysis, Validation, Investigation, Visualization, Methodology, Writing - original draft, Writing - review and editing; Kayla A Jara, Formal analysis, Validation, Investigation, Visualization, Methodology, Writing - review and editing; Susanne Bornelöv, Data curation, Software, Formal analysis, Validation, Visualization, Writing - review and editing; Marzia Munafò, Resources, Funding acquisition, Investigation, Visualization, Writing - review and editing; Vasileios Frantzis, Emma Kneuss, Investigation, Writing - review and editing; Elisar J Barbar, Formal analysis, Supervision, Funding acquisition, Validation, Visualization, Methodology, Project administration, Writing - review and editing; Benjamin Czech, Conceptualization, Data curation, Formal analysis, Supervision, Validation, Investigation, Visualization, Methodology, Writing - original draft, Project administration, Writing - review and editing; Gregory J Hannon, Conceptualization, Supervision, Funding acquisition, Methodology, Project administration, Writing - review and editing

## Author ORCIDs
Evelyn L Eastwood https://orcid.org/0000-0002-3932-0402
Kayla A Jara https://orcid.org/0000-0003-0406-2957
Susanne Bornelöv https://orcid.org/0000-0001-9276-9981
Marzia Munafò http://orcid.org/0000-0002-2689-8432
Vasileios Frantzis http://orcid.org/0000-0002-3497-700X
Emma Kneuss http://orcid.org/0000-0003-0662-8539
Elisar J Barbar http://orcid.org/0000-0003-4892-5259
Benjamin Czech https://orcid.org/0000-0001-8471-0007
Gregory J Hannon https://orcid.org/0000-0003-4021-3898

## Decision letter and Author response
Decision letter https://doi.org/10.7554/eLife.65557.sa1
Author response https://doi.org/10.7554/eLife.65557.sa2

# Additional files
## Supplementary files
• Supplementary file 1. List of all TQT-containing proteins enriched in Ctp IP-MS ($log_2FC > 2$ and p<0.01). The sequence of each TQT motif was identified using LC8Pred and is shown alongside gene names, number of motifs per protein and the corresponding LC8Pred amino acid score. Both TQT motifs in Panx are highlighted in green.

• Supplementary file 2. Thermodynamic parameters of Ctp-Panx interactions. Panx[TQT#1] refers to the construct in which the first TQT motif is mutated but the second site is intact (and vice versa).

• Supplementary file 3. List of siRNA sequences, qPCR primers, and fly stocks used in this study.

• Supplementary file 4. List of ChIP-seq, RNA-seq, and small RNA-seq libraries used in this study.

• Transparent reporting form

## Data availability
Sequencing data reported in this paper has been deposited in Gene Expression Omnibus (GSE160860). Mass Spectrometry data has been deposited to the PRIDE Archive (PXD022103 and PXD022105).

The following datasets were generated:

| Author(s) | Year | Dataset title | Dataset URL | Database and Identifier |
|---|---|---|---|---|
| Eastwood EL, Jara | 2021 | Dimerisation of the PICTS | https://www.ncbi.nlm. | NCBI Gene Expression |

| Author(s) | Year | Dataset title | Dataset URL | Database and Identifier |
|---|---|---|---|---|
| KA, Bornelöv S, Munafò M, Frantzis V, Kneuss E, Barbar EJ, Czech B, Hannon GJ | | complex via LC8/Cut-up drives co-transcriptional transposon silencing in *Drosophila* | nih.gov/geo/query/acc.cgi?acc=GSE160860 | Omnibus, GSE160860 |
| Eastwood EL, Jara KA, Bornelöv S, Munafò M, Frantzis V, Kneuss E, Barbar EJ, Czech B, Hannon GJ | 2021 | Dimerisation of the PICTS complex via LC8/Cut-up drives co-transcriptional transposon silencing in *Drosophila* | https://www.ebi.ac.uk/pride/archive/projects/PXD022103 | PRIDE, PXD022103 |
| Eastwood EL, Jara KA, Bornelöv S, Munafò M, Frantzis V, Kneuss E, Barbar EJ, Czech B, Hannon GJ | 2021 | Dimerisation of the PICTS complex via LC8/Cut-up drives co-transcriptional transposon silencing in *Drosophila* | https://www.ebi.ac.uk/pride/archive/projects/PXD022105 | PRIDE, PXD022105 |

The following previously published datasets were used:

| Author(s) | Year | Dataset title | Dataset URL | Database and Identifier |
|---|---|---|---|---|
| Fabry MH, Ciabrelli F, Munafò M, Eastwood EL, Kneuss E, Falciatori I, Falconio FA, Hannon GJ, Czech B | 2019 | piRNA-guided co-transcriptional silencing coopts nuclear export factors | https://www.ncbi.nlm.nih.gov/geo/query/acc.cgi?acc=GSE121661 | NCBI Gene Expression Omnibus, GSE121661 |
| Fabry MH, Ciabrelli F, Munafò M, Eastwood EL, Kneuss E, Falciatori I, Falconio FA, Hannon GJ, Czech B | 2019 | piRNA-guided co-transcriptional silencing coopts nuclear export factors | https://www.ebi.ac.uk/pride/archive/projects/PXD011415 | PRIDE, PXD011415 |

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
