## [Decision Letter]

**Acceptance summary:**

In this study Hannon/Czech and colleagues identify the dynein light chain Ctp as a new factor that co-purifies with the PICTS. Like other components of this complex, Ctp behaves like an effector protein that can mediate transcriptional silencing of a genomic locus. The important concept emerging from this study is the need for dimerization of the silencing complex via the identified novel component Ctp.

**Decision letter after peer review:**

Congratulations, we are pleased to inform you that your article, "Dimerisation of the PICTS complex via LC8/Cut-up drives co-transcriptional transposon silencing in *Drosophila*", has been accepted for publication in *eLife*.

In this study Hannon/Czech and colleagues identify the dynein light chain Ctp as a new factor that co-purifies with the PICTS. Like other components of this complex, Ctp behaves like an effector protein that can mediate transcriptional silencing of a genomic locus. The important concept emerging from this study is the need for dimerization of the silencing complex via the identified novel component Ctp.

Please take note of the points below.

Reviewer #1:

Minor concerns and suggestions that could improve the manuscript further:

1) Amongst the 2578 deregulated genes (RNA level) in ctp germline knockdown, are any PICTS or other piRNA pathway factors strongly deregulated as well? Can (additional) indirect effects be concluded unlikely?

2) It would benefit the openness of the study to reveal not only a list of enriched proteins for the Ctp coIP-MS experiment but the full dataset with the plotted enrichment and p values (Figure 4A)

3) Results paragraph two: here the text jumps too fast to include a broader readership. Why does a higher TE deregulation percentage compared to panx sh indicate a function in germline TE silencing for ctp?

4) I would tone down the confidence of the mechanistic conclusion based on the structural modeling (Figure 6D-E, subsection “Ctp drives PICTS complex self-association”) to underline that these are predictions rather than tested facts.

5) The authors speculate about a function in the retention of the silencing factors to chromatin (as was also speculated in the papers describing the PICTS complex) and also mention possible LLPS involvement. Such models could be made more useful to the field by discussing how they could be tested.

Reviewer #2:

Minor points for discussion.

1) It is definitely a surprising finding, given that Ctp has another life as part of the dynein motor. I am wondering how Ctp is able to organize assembly of specific complexes without creating non-productive complexes. For example, a scenario where Ctp homodimers interact simultaneously with Panx and other unrelated proteins that also have the TQT motifs. Such a complex would be a dominant-negative, as it sequesters both Ctp and PICTS components in non-productive complexes.

2) The authors demonstrate that Ctp is critical for PICTS function in a more direct way by promoting the dimerization of Panx (in the PICTS). This is done via Ctp interaction with Panx at two specific motifs on its C-term. Panx that fails to dimerize are inactive in a tethered repression assay. One conclusive experiment is where the authors artificially mediate Panx dimerization (with a leucine zipper domain) and this overcomes the need for Ctp for silencing by tethered Panx. I am wondering how a leucine zipper-mediated dimer would be same as that naturally mediated by Ctp? If the model proposed by the authors of LLPS formation (via increased RNA binding) is the pathway taken, then perhaps structural architecture is not so important.

3) In Figure 4E. Add Nxf2-interacting region of Panx to the cartoon.

---

## [Author Response]

Reviewer #1:Minor concerns and suggestions that could improve the manuscript further:1) Amongst the 2578 deregulated genes (RNA level) in ctp germline knockdown, are any PICTS or other piRNA pathway factors strongly deregulated as well? Can (additional) indirect effects be concluded unlikely?

We thank the reviewer for this comment. Of the components of the PICTS complex, only Panx expression was found to change significantly at the RNA level upon germline knockdown of Ctp, although the effect was small (log_2_ fold-change = -0.17). None of the established germline piRNA pathway factors were among the 2,578 deregulated genes and therefore we conclude it unlikely that this contributes to the observed transposon de-repression. However, it is difficult to separate direct and indirect effects in this experiment given the phenotypic severity caused by loss of Ctp in the germline. For clarity, we have provided additional tables summarising the differential expression analysis including the log_2_ fold-change and associated p-value for genes and transposons for all comparisons included in this manuscript as Figure 1—source data 1 and Figure 2—source data 1.

2) It would benefit the openness of the study to reveal not only a list of enriched proteins for the Ctp coIP-MS experiment but the full dataset with the plotted enrichment and p values (Figure 4A)

We agree with the reviewer. The data used to plot Figure 4A is already provided as Figure 4—source data 1. This table includes the log_2_ fold-change and associated p-value for all proteins detected in the Ctp coIP-MS experiment.

3) Results paragraph two: here the text jumps too fast to include a broader readership. Why does a higher TE deregulation percentage compared to panx sh indicate a function in germline TE silencing for ctp?

We thank the reviewer for pointing out this lack of clarity. We aimed to highlight that *ctp* knockdown results in TE upregulation and did not intend to directly compare the effects of both knockdowns, which is something we pick up in the Discussion (paragraph three). We have adjusted the text to improve the readability.

4) I would tone down the confidence of the mechanistic conclusion based on the structural modeling (Figure 6D-E, subsection “Ctp drives PICTS complex self-association”) to underline that these are predictions rather than tested facts.

While we would emphasize that this model is based on a number of known solved structures for Ctp homodimers bound to TQT peptides, we appreciate the reviewers concern and have adjusted the text accordingly to underline that these are predictions and hypotheses.

5) The authors speculate about a function in the retention of the silencing factors to chromatin (as was also speculated in the papers describing the PICTS complex) and also mention possible LLPS involvement. Such models could be made more useful to the field by discussing how they could be tested.

While this paper was in revision, Brennecke and colleagues reported that recombinant PICTS (which they call SFiNX) forms molecular condensates in vitro in a manner dependent on Ctp-mediated dimer formation (Schnabel et al., 2021). Future work will involve testing firstly whether this occurs in vivo, and whether it is essential for the silencing capacity of the complex. In this light, it will be important to identify mutants that separate the LLPS potential of the complex from the formation of the full complex and presence of all four components. We have added a comment about this to the Discussion.

Reviewer #2:Minor points for discussion.1) It is definitely a surprising finding, given that Ctp has another life as part of the dynein motor. I am wondering how Ctp is able to organize assembly of specific complexes without creating non-productive complexes. For example, a scenario where Ctp homodimers interact simultaneously with Panx and other unrelated proteins that also have the TQT motifs. Such a complex would be a dominant-negative, as it sequesters both Ctp and PICTS components in non-productive complexes.

We thank the reviewer for raising this interesting point and recognise that the vast number of Ctp binding partners requires highly specific formation of Ctp-dependent complexes. While we cannot rule out Ctp-mediated heterodimerisation, this has not previously been observed in other contexts. Such specificity can be achieved in a number of ways. Many known Ctp partners have additional self-association interfaces that are stabilised in the presence of Ctp. These favour the formation of homodimers rather than heterodimers of different Ctp partners. For example, binding of Ctp to Swallow results in homodimer assembly and the formation of a coiled coil adjacent to the TQT motif region (Kidane et al., 2013). Similarly, binding of Ctp to the dynein intermediate chain promotes self-association of a nascent helix domain (Benison et al., 2006). While loss of the TQT motifs has a significant impact on Panx self-association, it is likely that other regions of Panx form contacts in the dimer and contribute the specificity and stabilisation of the homodimer. We have added a sentence to the Discussion.

2) The authors demonstrate that Ctp is critical for PICTS function in a more direct way by promoting the dimerization of Panx (in the PICTS). This is done via Ctp interaction with Panx at two specific motifs on its C-term. Panx that fails to dimerize are inactive in a tethered repression assay. One conclusive experiment is where the authors artificially mediate Panx dimerization (with a leucine zipper domain) and this overcomes the need for Ctp for silencing by tethered Panx. I am wondering how a leucine zipper-mediated dimer would be same as that naturally mediated by Ctp? If the model proposed by the authors of LLPS formation (via increased RNA binding) is the pathway taken, then perhaps structural architecture is not so important.

We thank the reviewer for their suggestion, and agree that while only speculative, it is possible that the structural architecture of the dimer interface might not be so important for the silencing function of the complex. To test this hypothesis, the leucine zipper could be fused at different positions within Panx, since in the experiment presented here, it was fused to the Panx C-terminus to mimic as much as possible the wildtype dimer assembly. However, in the absence of a structure of Panx in complex with its partners, we risk disrupting other essential structural elements and binding interfaces. We note that the 2xTQT mutant is functional in the DNA tethering assay, where the N-terminal LacI fusion effectively acts like the zipper in being an alternative self-association interface. However, whether this reflects the fact that N-terminal dimerisation is sufficient to rescue silencing function or instead that dimerisation is not a requirement for silencing in the DNA tethering assay cannot be easily deduced. Overall further investigation of the silencing mechanism will be required to address these ideas thoroughly.

3) In Figure 4E. Add Nxf2-interacting region of Panx to the cartoon.

The region sufficient for Panx’ interaction with Nxf2 was indicated as “CC2” (details in Fabry et al., 2019). For clarity we have added this information to the figure legend of panel E.